## [Transparent Peer Review file · Nature Communications]

Combinatory differentiation of human induced pluripotent stem cells generates functional thymic epithelium driving dendritic- and CD4/CD8 T-cell development.

Corresponding Author: Dr Matthieu Giraud

Version 0:

Reviewer comments:

Reviewer #1

(Remarks to the Author)

Giraud and colleagues set out to develop an improved protocol for generating thymic epithelial cells from human iPSC cells. To do this, they used a Plackett-Burman combinatorial screening approach to test the requirements for different growth factors, selected based on studies of thymic organogenesis and previous attempts to differentiate hESC into thymic epithelial cells and assayed using bulk RNAseq of the cells produced in each condition with subsequent comparison to existing pharyngeal endoderm and thymus development datasets. The authors used a four step protocol for this analysis (hiPSC to definitive endoderm (DE); DE to anterior foregut endoderm (AFE); AFE to third pharyngeal pouch endoderm (3PPE); 3PPE to thymic epithelial progenitors (TEP). As step 1 (hiPSC to DE) protocol is already well established, the authors focused on steps 2-4. Using their data in conjunction with knowledge from the literature, selected an optimised set of growth factors for each of these steps. To characterise the cells produced in further detail they then performed further bulk RNAseq at a series of timepoints across the optimised differentiation protocol. Subsequently, the hiPSC-TEP produced in the optimised protocol were tested in two functional assays – a 2D co-culture system, and a 3D 'organoid' model. Overall, the approach taken was interesting and based on sound rationale (although this appeared not to be applied consistently in the choice of factors), but the outcome was rather disappointing. While epithelial cells that expressed some markers characteristic of thymic epithelial cells were clearly produced it is not clear from the data presented how well (or whether) these cells could support T cell development (since the required controls are not shown), nor how well they represented the epithelial subsets present in the human thymus (indeed while the data show that mTEC-like cells may have been generated, they do not provide convincing evidence for the generation of mature cTEC). This diminishes enthusiasm for the manuscript. Therefore, although credit is given for the rational approach taken and the authors are applauded for presenting a more in depth analysis than many of the papers recently published in this area, it is not clear that the current work represents a major step forward over the best currently published protocols (e.g. Ramos et al 2023).

Specific comments are provided below:

General

- Throughout, the clarity of the manuscript could be improved, it is rather confusingly written in several places. Typos should also be tidied up including in the figures.
- Please show negative controls for all immunofluorescence staining and also for the flow cytometry analyses presented, along with the full gating strategy for the latter (these can be provided as supplementary figures).
- Please make sure that n is stated for all analyses, including immunofluorescence.

Specific comments

Fig 1: The approach taken is appropriate (and welcome) and appears to have been applied well. Overall, the differentiation protocol shares similarities with previously published protocols. There are some key differences though (e.g. activation or inhibition of BMP signalling in the DE-3PP and 3PP-TEP stages, etc) and discussion of these differences would be appreciated, as would a summary of the precise optimised differentiation conditions (in the text and in Fig1e). A weakness is that, although the aim was to be guided by the outcome of the combinatorial analysis, some factors appeared to be added to the 'optimal' medium for each set because they were already included in established protocols, even when the current

analysis suggested no benefit.

Fig 2: The data presented in this figure assess how well the hPSC-TEP/TEC phenocopy primary human TEC. However, in most of the modules highlighted, there appears to be little correspondence with primary TEC (Fig 2). The fitting to the Magaletti data suggesting cTEC identity is interesting (Fig 2c), if surprising based on the heatmap data. However, the authors should provide details of the parameters used to allow superimposition of the two datasets in order that these data can be properly evaluated by the reader.

Figure 2a:

- Please provide more information as to how the modules were identified.
- PSMB11, PRSS16, HLA-DR, CD80, CD86 are mentioned in the text but not annotated on the figure, please correct this omission.
- What does the un-annotated module (above 'EPCAM' module) represent?
- Why are the genes in the last module so over-represented?
- These data do not convince that the protocol has developed TEC, although the lack of some canonical TEC markers in the primary TEC dataset suggests there may be a problem with those data. The hiPSC differentiation data could be checked against human TEC data in the public domain (e.g. from the Teichmann lab).

Fig 2c:

– please provide details of how the deconvolution analysis was done i.e. what parameters were used to fit the bulk RNAseq data onto the Magaletti data? If I understand correctly, the Magaletti data are scRNAseq, whereas the dataset generated herein (for Fig 2) is bulk RNAseq. No details appear to be provided as to how these datasets were merged, again making it hard for the reader to evaluate their significance.

Related, SF5: The staining with FOXP1 looks real, the PAX9 stain is OK though less convincing, however, I find it surprising that all cells stain with these markers. Please provide negative control staining for both antibodies.

Figure 3

Fig 3a - please clarify whether the controls are unstained, or FMOs.

Fig 3b - the CD205+ TEP do not appear to resemble TEP or TEC very closely, but appear to strongly express a signature associated with mTEC fate. Please show the expression patterns of cTEC associated genes (especially since CD205 is cTEC affiliated later in development), and comment on these in the text. Also relevant here is line 296 – the Krt19+ population at E12.5 is a separate mTEC-fated progenitor, not a bipotent TEP, as shown by Lucas et al and also by Farley et al 2023, and evident in Magaletti etc, while the text in line 296 rather suggests that it is part of a common TEPC signature.

Fig 4: Here, the authors set out to push the PSC-'TEC' towards the mTEC fate using RANKL, as well as IL7, SCF and FTL3L (Of note, it is well established that in vivo RANKL does not regulate the cTEC/mTEC fate choice but allows mTEPC to expand).

- The 'ETP' population used for the co-culture is Lin-CD34+CD7+ (lin= CD3 CD4 CD8 CD14 CD19 CD56), and, from the analysis shown in supplementary figure 6 a and b appears to contain some cells (~10%) that are starting to express CD4, and some cells that express CD3. Based on this, it seems more appropriate to call the sorted population 'DN' rather than ETP – although the majority of the CD4- fraction of sorted cells appear to be DN1, there are clearly other cells in the sorted population including cells that may have passed the beta selection checkpoint. Please clarify this in the text.
- The authors observe association of the 'ETP' with TEC, and expression of K14, K5, MHCII and Cldn3. They state that this phenotype represents a fraction of the differentiated cells, but from the data presented it looks like most of the differentiated cells express these markers. Please clarify the proportion of cells in each dish/on each slide which are positive for the above markers.
- Fig 4 does not examine what happened to the 'ETP' in the co-culture, did they differentiate? Please comment on this in the text and show related data.

Figure 5

- It seems somewhat surprising that the authors choose to use a set of markers of 'TEP' (CDH10, ACKR3, TBX3 and TGFB2) that does not include any of the markers normally taken to characterise the TEP populations present in early development (or indeed the various TEP populations suggested in the adult thymus). Inclusion of some commonly agreed TEP markers in this analysis would be welcome.
- It is not clear that the Monocle analysis adds much useful information, the authors could consider moving these data to a supplementary figure.
- It is not clear whether in Fig 5f, the cluster 1 and 2 cells are part of a population that is present in normal human TEC, or whether they (and indeed the cluster 3 cells) are clustering separately but adjacent to the ex vivo TEC populations. It appears to be the latter, as all/almost all of the cells in those areas represent the hPSC-'TEC'. Please clarify this in the text, not least as it affects the strength of the conclusions that can be drawn.
- The data shown in Fig 5f (from d7 organoids) do not strongly support the notion that bona fide TEC have been made in this protocol (rather, they suggest that cells that have some resemblance to immature cTEC have been made, but that these have not generated normal differentiated TEC populations. A few cells resembling but not identical to mTEChi cells are also present). Please comment.

Fig 6

- See Line 398 – that statement that thymocytes were concentrated around the projections does not appear to be strongly supported by the data shown. Rather, thymocytes appear scattered through the space examined (see Fig 6b, lower left

panel; these are d28 data). Please provide quantitative data confirming a positive association, or tone down this comment.

- Line 402 – please bear in mind that KRT5 does not necessarily indicate mature epithelium. There are plenty of examples (including in fetal thymus) where it marks stem or progenitor cells. The text should be adjusted accordingly.
- In Fig 6e, without knowing more details of how the deconvolution was done, it is hard to know how significant these data are. Please provide this information. It looks as though the hiPS-‘TEC’ share features with cTEChi, and with some minor mTEC populations, but not with the major mTEClo and mTEChi populations.
- A weakness of the bulk RNAseq approach is that it is not possible to determine the phenotype of single cells or the proportional representation of ‘TEC-like’ cells within the organoid. In view of this (see also comments above), the conclusion drawn (line 439) is overstated. Please rephrase to avoid overclaiming.

Fig 7

- Please show the negative controls for the flow cytometry stains, as a supplementary figure (preferably FMOs)
- Panels a and c – the levels of CD3 look consistent between weeks 3 and 5 in panel a (and fairly low), but are much higher in week 3 in panel 5. Please comment.
- Please show CD4 v 8 profiles at timepoints earlier than week 5 (as FACS plots, not just a bar chart). Please comment on the lack of double positives at week 7 (panels d and e) - to my eyes, there are none in the plot shown in panel d or the upper plot in panel e, and very few in the lower panel e plot. I realise these are gated on CD3 but nonetheless I would expect to see DP in this population. The plots shown appear to contradict the bar chart, in which DP is the predominant population at week 4 and week 5. In line 456, the authors state that most CD3+ cells are DN, but that ‘CD4+CD8+ DP, CD4+CD8- SP4 and CD4-CD8+ SP8, albeit in lesser proportions (Fig. 7d)’. However, no DP are present in Fig 7d. This statement also appears to contradict the data shown in Fig 7f. If the input CD45+ population had been pure ETP, this would not be a concern, but since 10% of cells in that population are starting to express CD4 (and some of which express CD3), I am concerned that the SP populations present may reflect in vitro maturation of already committed DNs (including some that may already have passed beta selection) driven by the cytokines provided. Please show representative CD4 v CD8 plots at each timepoint after gating on CD45 only, and also show the negative control data (i.e. culture of the DN in the absence of the hiPSC-TEP, in the same medium as the organoids, related to SF7b).
- Line 451 – can the authors really say commitment here, as they really start with DN rather than ETP (some of the ‘ETP’ will already have undergone commitment). Please modify the text as needed. Related, please show thymocyte subset data for the controls (SF7b) rather than just cell numbers; these data are essential.

Minor comments:

Line 74 – The literature on TEPC has moved on a bit from these two references, please update. In particular, whether there is a common TEPC that plays a major role in generating or replenishing TEC remains rather open.

Line 76 – The Notch signals described in these papers include Notch signalling that is active pre-haematopoietic colonisation, so can’t be provided by cross-talk. Please clarify the text accordingly.

Line 90 – Please make sure the referencing for the directed differentiation approaches is complete.

Line 269 – its not clear at what point, if ever, in early development there are bipotent progenitors, though both cTEC and mTEC progenitors express b5t (and presumably, CD205). See Magaletta et al 2022, Farley et al 2023, Liu et al 2020, Li et al 2020, Nusser et al 2022 among others. Please correct this statement to accurately reflect the literature.

Line 313 – the authors say ‘in addition to a cortical one’ [fate] but to my mind, differentiation into a cTEC fate has not been adequately demonstrated, by the authors admission the cells generated lack canonical cTEC markers. Please correct accordingly.

Line 330 – The data presented show that something like an mTEC may have been generated, though no details of which subtype(s) of mTEC are provided. There does not appear to be any convincing evidence that mature cTEC are generated. Please modify the statement on line 330 to reflect this.

Supplementary Figure 1d: please state in legend what 1 and -1 mean, I assume it means hi and lo dose but this is not clear – were factors also omitted?

Reviewer #2

(Remarks to the Author)

Reviewer #3

(Remarks to the Author)

The creation of in vitro systems that fully recapitulate thymic stromal features/function is essential for regenerating the thymus in pathological settings and in aging. However, capturing and maintaining the complexity of the thymic microenvironment in vitro from iPSCs, especially the various types of thymic epithelial cells (TECs), has been a challenge despite numerous attempts by several groups. Moreover, while recent studies reported in vitro systems capable of fully

reproducing thymocyte development, some specific steps of thymocyte selection/maturation remain an obstacle, raising the need for a fine-tuning of current thymic organoid models.

In this report, Provin and colleagues take on the challenge and use an original methodology to rationalize the culture conditions required for differentiation of iPSCs to TEPs, coupling statistical-mathematical approaches to previously published gene expression datasets on early thymic organogenesis, to recreate optimal experimental designs. Focusing on 4 key steps of thymic organogenesis the authors were able to confirm/demonstrate which specific pathways previously used in the field were needed for the induction of TEP fates from iPSC. The authors also identified a new pathway (IGF-1) to be targeted in vitro for TEP induction. In addition, they showed the higher efficiency of their protocol for TEP generation in comparison to two previously published reference methods, and its reproducibility across iPSC lines.

After extensive phenotypic and transcriptomic characterization of iPSC-derived TEPs, they proceeded to test their potential to further differentiate into TEC subsets in vitro, by aggregating cells with CD34+CD7+ human thymic progenitors to create human thymic organoids (hTOs). They briefly assessed the phenotype of TECs in later stage hTOs and show the possible presence of mature mTEC subsets. This maturation process was obtained in vitro, bypassing the need for in vivo transplantation of organoids usually described in the field. Finally, the authors identified CD8SP and CD4SP and dendritic cell subsets derived in hTOs.

This work is original in its DOE approach although this analysis relies on previously published protocols. While the methodology for deriving TEPs and hTOs has some novelty, the evidence showing this system can generate sufficient and functional T cells in a way that surpasses current models in the field is limited.

Below is a detailed list of major and minor comments/points to be addressed:

Major comments:

- There is a lack of quantitative data throughout the manuscript: how many TEPs can be obtained from iPSCs? How many cells are needed (TEPs and ETPs) to make hTOs? How many hematopoietic cells (T, DC) can be harvested from hTOs (the FACS profiles suggest this is very limited)? How reproducible is the hTO system in showing T cell output from TEPs?
- Clarification of what is different from other published protocols using the DOE revised methodology is essential to evaluate what is new in the paper.
- Figure 7: the authors mention the presence of all thymocyte subsets in hTOs, but there is no data on the kinetics of T cell development from early to late time point in hTO cultures including the development of precursors, most importantly DPs. CD4 versus CD8 flow data would be an important addition to the bar graph in Figure 7f.

The authors describe the presence of mature T cells generated in the hTOs, based on the expression of CD3+ cells at selected time points but TCRab expression data is very weak. CD3 is also expressed by NKT cells so additional markers should be used to discriminate between T and NKT cells. What are the CD3+TCRa/b- cells? Moreover, a better characterization of the maturation of SP4/SP8 cells in hTOs is needed (CD3 vs TCRab expression) at all stages of hTOs development.

- While the authors provide some phenotypic characterization of TEP, a more detailed phenotypic analysis of other stromal cells in 3D conditions (hTOs) throughout the length of the cultures is missing. In particular, the frequency of EpCAM+ and/or EpCAM+CD205+ cells between D0 and D35 in hTOs is not shown. Is FOXN1 detected at early/mid/late stages of hTO cultures?
- Is there any spatial evidence of distinct cortical/medullary areas in hTOs?
- Thymic cross-talk between epithelial cells and thymic progenitors/T cells has not been demonstrated and without stronger data this term should be deleted from the title and conclusions.

Essential experiments/data

- Quantification of TEP and hTO cell output (stromal and T cell) is essential to show the system is superior to others in the field.
- Stronger evidence of thymopoiesis including CD4+CD8+ development and kinetics of T cell differentiation is required as described above.
- ETPs are a heterogeneous population, containing cells already primed for T-cell fate in the thymus. Of note, the cited paper Lavaert et al, 2020 defines the earliest thymic progenitors (TSP1) as CD7- (by protein) and only CD7+ by RNA, so the authors have seeded TSP2 cells into the cultures, a population that has already committed to T cell lineage. Evidence that the TEPs produced from iPSCs in this system can support T cell development from less mature progenitor sources (cord blood CD34+ for instance) is required.
- Flow cytometry data showing CD205 and HLA-DR expression in hTO TECs is unclear; a better characterization of the stromal compartment of hTO (and cellular compartmentalization) should be achieved by immunofluorescence.

- Immunofluorescence experiments would better demonstrate the cross-talk between stromal and hematopoietic progenitor/cells instead of the current data shown in Figure 4.
- DLL1 and DLL4 expression are key markers of TECs. RNA-Seq expression of these are relatively weak in CD205+ TEPs whereas Jag1 and 2 expression appears to be more prominent (Fig 3b). Protein expression of these key markers (at least DLL4) should be given with appropriate controls.
- The statement (line 258) that FOXN1 expression showed a complete differentiation into TEC-fate (Fig S5B)" is misleading. Only a few cells are shown and no summarized data on the % of FOXN1 cells in culture is given. In addition, while FOXN1 is necessary for proof of fully differentiated TECs it is not sufficient.
- Figures throughout need more complete explanation in legends.

Minor comments:

- Discussion: hTOs are said to contain fibroblasts, but there is no evidence showing the presence of fibroblasts in the Results section. Mesenchymal support of TECs would be expected to be required for maintenance of the hTOs. Where do the fibroblasts originate from?
 - Co-culture of ETPs was used to help mature the TEPs in vitro-discussion of how/if this would be used practically for generation of TECs for clinical use would be helpful.
 - Are there any regulatory T regs in hTO with CD4+ T cells?
 - Inconsistent time point labeling: some figures specify D35, others W5.
 - Introduction: line 54-57 needs citations for the human TEC markers cited
 - Line 269: the statement that CD205 marks TEP that are bipotent should be clarified as CD205 is not specific -the ref for this should be cited
 - Careful editing of grammar and meaning is needed throughout: the use of "remarkably" should be kept to a minimum if used at all.
- some examples in the Intro:
- Line 77: Understanding of regulationhas shown progress
 - Line 86-understanding of regulation...has shown progress
 - Line 100-this is confusing as most forms of SCID are due to mutations in hematopoietic cells and so will not benefit from TEC replacement
 - Line 105-confusing example of application of TECs

Reviewer #4

(Remarks to the Author)

Reviewer #5

(Remarks to the Author)

In recent years, many techniques have been reported to induce differentiation and functional maturation of TECs. However, the myriad of protocols has left a need for systematic testing of combinatorial effects of different factors to induce differentiation of TECs at early stages and to establish an in vitro maturation protocol without grafting of TEPs in vivo. The current work systemically investigates the effects of many factors at three early stages (from DE to AFE to 3PPE to TEP) on the differentiation of human iPSCs into TEPs as well as the maturation of cortical and medullary TECs in hydrogel-based 3D co-cultures with primary hematopoietic progenitors ETPs. The authors could also show that ETPs differentiate into CD4 and CD8 T cells and dendritic cells in the 3D hTO co-culture system. The experiments are well designed and the results add to the overall incremental understanding of the factors to differentiation induction at the different transition stages. Impressively, in the 3D hTO co-culture system, the iPSC-derived TEPs continue to mature into functional TECs that support human ETPs to differentiate into T cells and dendritic cells.

Some comments are provided to improve the manuscript.

1. Fig. 1d and Supplementary Fig. 1d: It would be interesting to know how the authors used Plackett-Burman designs as a screening method to evaluate the most significant factors with the fewest experiments. For example, why is it necessary to have 36 runs for 5 factors on two levels (AFE: D5 – D7)?
2. “However, we chose to keep BMP4 and FGF8 because of their consensual use to achieve 3PPE differentiation in previous studies^{37,39,40}.” “Similarly to above, we chose to keep FGF10 and EGF since we observed higher growth rate in cultures supplemented with EGF.” Given this, some of the key findings would warrant testing in a combination without these factors at the transition stages of 3PPE (D7 to D11) and TEP (D11 to D131) to determine whether they really add any significant benefit or can be eliminated to simplify the protocol.
3. Lines 291-293: “We also identified the NOTCH ligands DLL1, DLL4, JAG1 and JAG2 whose coding genes show robust higher expression in TEP vs iPSc (Fig 3b, Right)”. Do the authors mean in CD205+ TEP vs TEP (or iPSC)?
4. Supplementary Fig. 6a: Please check the method section, should keep consistent, either using CD4/CD8 depletion or CD3 depletion, or both?
5. Supplementary method section is missing, please provide information on antibodies used in the study, also information and concentrations of factors used for TEP aggregation with ETPs and 2D co-culture, including RANKL, FGF10, IGF1, EGF, SCF, IL7, and FTL3L, lines 637-639.
6. 2D co-culture and 3D hTO co-culture: please provide the ratio and cell numbers of TEPs and ETPs used.
7. Fig. 4: b, what are differences between left and right images? c, How is the co-localization quantified? d, What percentage of TEPs become mTECs?
8. Fig. 4b, 5b, 6a, S1: please insert scale bars.
9. The discussion does not compare their hTO co-culture results to previously published work with grafting of TEPs in vivo for maturation induction of TECs and efficiency to support T cell and dendritic cell development.

Version 1:

Reviewer comments:

Reviewer #3

(Remarks to the Author)

In the previous version of the manuscript “Combinatory differentiation of human induced pluripotent stem cells generates functional thymic epithelium driving dendritic- and CD4/CD8 T-cell full development”, by Provin, d’Arco, Giraud et al., we identified major points to address to justify statements made by the authors.

In this new revised version of the manuscript, the authors made significant improvements pertaining to the major suggestions for review, notably:

- Adding quantitative information (cell numbers) generated from iPSC (TEPs) and T-cell population differentiated in hTOs throughout the manuscript;
- Detailing the number of TEPs and progenitor cells needed for making hTOs;
- Clarifying the use of the DOE method;
- Verifying the reproducibility of the hTO by increasing the number of experiments;
- Adding data on the kinetics of T-cell development;
- Testing the efficiency of the hTOs in generating T cells when starting from HSPC not primed towards T-cell fates;
- A better characterization of TECs in hTO by showing gene and/or protein expression of markers of TEC identity and function (notably mTECs);
- A better characterization of non-T hematopoietic populations (DCs) and non-TEC stromal cells in hTOs;
- Confirming hTOs do not generate cells with NK and NKT fates;
- Improving figure legends, texts and revising statements based on new experiments.

Our minor points were also considered and addressed in this revised manuscript.

Two main issues remain in the current version

1. There is no functional analysis of the T cells produced, likely because of low cell yield. The authors’ statement that this will be solved using future changes in methodology is rather overconfident and the issue should be directly addressed in the discussion as to why the output is so low. This is also an essential hurdle to overcome before translation is feasible and will also limit the experimental usefulness of the model.

A major point that should be corrected throughout the manuscript is of the nature of the seeding hematopoietic cells that can generate T cells in the hTOs. The only hematopoietic progenitors (whether from CB or iPSCs or thymus) able to generate T cells in the hTOs are those that express CD7 (i.e. CD34+CD7+ cells). As mentioned by Reviewer 1, CD34+ CD7+ cells are not HSCs nor true thymic ETPs as described by the authors. In CB they are mostly lymphoid-committed with DC and NK potential. In the case of thymic progenitors, CD34+CD7+ cells are largely T committed lineage. True ETPs with multilineage potential are CD34+CD7- (Le et al, Immunity 2020 doi: 10.1016/j.immuni.2020.05.010). The inability of the hTOs to generate T cells from CD34+CD7- cells suggests that critical factors required for induction of T cell specification and commitment (most likely Notch signaling), present in normal thymus, are missing in the hTO microenvironment. This deficiency in the system is also seen in some (although not the OP9DLL4/4 or ATO) models and should be clearly acknowledged as such in the manuscript.

Reviewer #4

(Remarks to the Author)

Reviewer #5

(Remarks to the Author)

The authors have addressed all of my previous comments; however, there are still some issues that require further clarification. In particular, the authors have added new experiments to support their conclusions.

1. Please avoid using the same abbreviation, DE, for both "definitive endoderm" and "differentially expression" in the manuscript, as this may cause confusion. Additionally, please carefully review the manuscript including the figures for grammatical errors and typos, for example, "differentially expression", "characterization", "PAX9 et FOXN1", "Protocol reproductibility", "An 3D thymic", "FTL3L", "reagregation", "HLA-DRhi et lo", "pharyngeal"
2. Please provide the gene lists for the different reference cell populations used in the comparison of iPSC-derived samples to the two atlases. Additionally, include the GEO accession codes for the DGE-seq data from iPSc to TEP.
3. Supplementary Fig. 4b: Are the proliferating cells in the differentiation end product TEPs or another cell type? How is the TEP population defined here? What percentage of the differentiation end product are TEPs? Considering Fig. 2a and Supplementary Fig. 6 (modules 4 and 6), a set of genes related to muscle development appears to be highly expressed in the differentiation end product (D14-D22). Could you provide an explanation for this observation?
4. Supplementary Fig. 5b: It would be informative to include a comparison between primary TECs and D14-D22 iPSC-derived samples.
5. What percentage of cells are double positive for FOXN1 and PAX9 (Supplementary Fig. 7a) in the differentiation end product?
6. Fig. 3c and line 334: The manuscript states that "one third of these cells with inferred TEC signature are medullary oriented", how was this proportion calculated? In Fig. 3c, the fraction appears to be less than 20%.
7. Fig. 3d and line 335: The manuscript states, "Moreover, classic cTEC markers are less expressed in the CD205+ TEPs". Less expressed compared to what population? This statement is unclear and needs further clarification.
8. Lines 336-337: the manuscript states, "identified NOTCH ligands DLL1, DLL4, JAG1 and JAG2, whose coding genes exhibit robustly higher expression in TEPs compared to iPSc (Fig. 3b, Right)." However, in Fig. 3b (Right), these genes appear to be expressed at lower levels in TEPs, with the exception of JAG2. As noted in my initial review, the authors may be referring to CD205+ TEPs, but in this case, DLL1 expression appears comparable to, or even lower than that in iPSCs, not higher. This should be clarified.

Reviewer #6

(Remarks to the Author)

While the authors have been largely responsive to reviewer 1's concerns, there are still a few questions that remain, especially about the maturation and development of thymocytes within the organoid system. This group has done a commendable job on generating what look to be organoids that resemble thymic epithelium in many ways. The major claim of this manuscript is that unlike other approaches that require an in vivo transplantation step, this is a fully ex vivo organoid able to generate mature T cells. However, I am not yet fully convinced that is what they are doing.

Firstly, CD4/CD8 expression seems to follow a curious pattern. On Line 566-568 the authors claim that there are CD4+CD8+ double positive cells (Fig. 7D) but that is not apparent from the flow plots being presented (a point that was raised in the prior reviewer comments). In Fig. 8 there are more DP cells, but as far as I can gather the reason why is that in Fig. 8 there is a broader CD45+ gate, not excluding CD3- cells. Based on this it seems that most DP cells are CD3-. This is surprising given the proportion of CD3+ cells that seem to be generated. In Fig. 8a it seems that there are a lot of both CD4+ and CD8+ single positive cells that are also CD3- (and very few that are CD3+). Finally, it also seems that most CD3+ cells are not TCRab or gd+. Based on the argument that the abundant CD3+TCR- cells are DN cells expressing preTCR (which would explain the expression of CD3 even early in the cultures), this would suggest downregulation of CD3 between the preTa stage and DP (and later) stages? Ideally, characterization of these earlier DN stages of T cell development using markers such as CD34, CD7 and CD5 would reveal what is happening within the DN compartment with respect to this unexpected expression of CD3. There also seems to be significant variation in outcomes with Fig. 7a showing 17% CD3+ at day 21 but Fig. 7c

showing 47% (this is reflected by the graph in Fig. 7b which shows a very large spread of CD3+ cells at day 21). Is there a difference in TCR expression in the very high CD3 expressing organoids vs the low CD3 expressing? In Fig. 7D the authors would like to claim differentiation to a mature CCR7+CD62L+ phenotype, but are these cells TCR+ or just CD3+? Ultimately, what evidence do the authors have that there are a significant fraction of CD3+ cells have actually undergone TCR recombination with selection? RAG induction? CD5 signaling? TRECs? Are they generated but is it just an extremely inefficient process? The biggest innovation of this study is the ex vivo generation of mature T cells in thymic organoids – even if in small numbers to start - so ensuring the robustness of this claim is important. Furthermore, the authors do not seem to have fully addressed these same questions about TCR expression from Reviewer 3 in their response.

In Fig. 8c the authors present kinetic data but there is no change in proportion of SP4 or SP8 cells from a very early timepoint of day 10. Do the authors think there are mature T cells that early and then do not change in proportion (despite the fact that no new hematopoietic progenitors are being seeded) across the timecourse of the organoid? This does not seem to fit with what we know about T cell development kinetics in the thymus and what could be expected to happen to SP proportions once earlier precursors are depleted.

Finally, I would suggest that the authors could again spend some time working on the wording in the manuscript (including greater experimental detail in the figure legends) to make it easier to follow. In addition, while I applaud the inclusion of a lot of validation data, 22 supplemental figures seems excessive and adds to the complexity. Including only the truly pertinent supplemental data may assist in the flow of the manuscript.

Version 2:

Reviewer comments:

Reviewer #3

(Remarks to the Author)

Most important issues are now addressed. However I continue to believe that the conclusions of how well the hTO can recapitulate normal thymopoiesis should be more circumspect. Given that the hTO model was only shown to generate T cells from primary thymic or iPSC-derived CD34+CD7+ cells, and not CB CD34+CD7- cells it is clear that T cells cannot be produced from uncommitted HSC or HSPC in the hTO system (unlike other in vitro systems). This very likely because of insufficient notch ligand expression in the iPSC derived epithelial cells, possibly because of immaturity. The poor output of CD4+CD8+ (DPs) also suggests that T cell differentiation is not proceeding normally (as suggested by reviewer #6). The statement (line 776) that “T-cell output remains modest mainly because the number of hematopoietic progenitors available from primary thymic samples restricts input and thus limits the size of organoid batches” begs the question of why the hTO system only worked with thymocyte progenitors.

While an alternative approach to produce TECs from iPSCs is a valuable addition to the literature I believe that it is important for the readership to understand the limits to the conclusions that can be made from the data (modifications should be made for example: line 663 “...functional thymic organoids supporting full thymopoiesis in vitro”)

I would also suggest removing the new sentence at the end of the abstract (line 41-42) “...generating mature thymic immune cells of clinical relevance and, through optimization of their production, paving the way for future cellular therapies.” -It would seem that the clinical relevance of the cells and their optimization of production are yet to be explored and should be made clear are speculative at this point.

Reviewer #4

(Remarks to the Author)

Reviewer #5

(Remarks to the Author)

In the new revised manuscript, the authors have thoroughly addressed all of my comments, I have no further questions or concerns.

Reviewer #6

(Remarks to the Author)

Let me begin by saying that I like the approach the authors have taken and they have done a commendable job to advance the differentiation of functional TECs from pluripotent cells. I also appreciate the authors have tried to do as much as they can with their existing data to answer my questions and concerns – and I do genuinely try to avoid asking for experiments where they are not always needed (especially having been brought in after another reviewer dropped out). However, I am still concerned that the data presented are not entirely convincing that there are truly TCR rearranged and functionally

expressing T cells being generated (let alone being MHC restricted and functional). Ideally, I would have loved to see additional data such as TRECs, dual staining for both CD3 and TCR, and even a functional measure of TCR engagement; however, I appreciate that may be beyond the scope of the manuscript and do not feel that these concerns should hold up publication.

Further digging suggests that, unlike mouse equivalents that typically detect only single TCR chains, the TCR antibodies used do seem to bind both alpha and beta chains (or gamma and delta chains) so should be detecting fully rearranged TCR (and not pre-TCR rearranged beta chain for instance). This should be highlighted as evidence of full TCR rearrangement. Progress has certainly been made in these systems, and it would be beneficial for the community to have these studies published, but ultimately they suggest that there is extremely inefficient generation of mature T cells.

I appreciate that the discussion does acknowledge a likely bottle neck between DN and DP, but to address my concerns, I suggest that the authors should consider working on some of the language to more carefully describe what is being generated and the limitations as they stand. For instance, the title referencing "full T cell development" seems a little premature since there is no functional evidence that these are indeed fully mature T cells. In the abstract, perhaps the language could be softened (further) to suggest that further optimization will be needed as the generation of mature T cells is extremely inefficient and the cells generated have not yet been tested functionally.

We would first like to thank the reviewers for their insightful and constructive comments, which have allowed us to significantly improve our manuscript. We are particularly grateful for the recognition of the rationale behind our Design of Experiment (DoE) approach and the analyses performed. We provide a detailed point-by-point response in blue for each reviewer below. We have highlighted changes to the revised manuscript using tracked changes in red.

MAIN REVISIONS INCLUDE:

* **Enhanced immunofluorescence characterization of iPSc-derived thymic epithelial progenitors (TEPs).** To confirm the identity of the cells generated by our DoE-based protocol, we performed additional immunofluorescence experiments. These new data demonstrate that iPSc-derived TEPs express key NOTCH ligands (DLL1, DLL4) and thymic epithelial markers (FOXP1, PAX9), with appropriate negative controls to validate specificity (**new Fig 3e, revised Supp Fig 7a**).

* **Improved characterization of TEC maturation dynamics in 2D and 3D systems:** We now provide a more comprehensive analysis of the maturation status of TECs derived from iPSc, in both 2D co-cultures and 3D human thymic organoids (hTOs) (**new Fig 6g, new Supp Fig. 10, 12-13**).

* **hTO experiments with detailed time-course analysis.** We have thoroughly reworked this section and now show a total of 11 independent hTO experiments (**including newly conducted replicates**) with **comprehensive flow cytometry gating strategies and negative controls**. The detailed characterization demonstrates consistent generation of single-positive CD4 and CD8 T cells as the organoids mature (**new Fig 8**).

* **New hTO experiments using iPSc-derived hematopoietic stem and progenitor cells (HSPCs).** To further validate the capacity of hTOs to support T cell commitment and differentiation, we performed a new series of experiments using iPSc-derived HSPCs. Specifically, we seeded hTOs with uncommitted CD34⁺CD44⁺CD7⁺ HSPCs, and demonstrated their successful differentiation into TCR $\alpha\beta$ -positive single positive T cells, along with CD14⁺ myeloid cells comparable to ETP-seeded hTOs (**Supp Fig 19-22**).

* **Significant revision and clarification of figure legends.**

* **Careful editing of the manuscript for language**, including grammar and typos.

Comments to Reviewers

Reviewer #1 (Remarks to the Author):

Giraud and colleagues set out to develop an improved protocol for generating thymic epithelial cells from human iPS cells. To do this, they used a Plackett-Burman combinatorial screening approach to test the requirements for different growth factors, selected based on studies of thymic organogenesis and previous attempts to differentiate hESC into thymic epithelial cells and assayed using bulk RNAseq of the cells produced in each condition with subsequent comparison to existing pharyngeal endoderm and thymus development datasets. The authors used a four step protocol for this analysis (hiPSC to definitive endoderm (DE); DE to anterior foregut endoderm (AFE); AFE to third pharyngeal pouch endoderm (3PPE); 3PPE to thymic epithelial progenitors (TEP). As step 1 (hiPSC to DE) protocol is already well established, the authors focused on steps 2-4. Using their data in conjunction with knowledge from the literature, we selected an optimised set of growth factors for each of these steps. To characterise the cells produced in further detail they then performed further bulk RNAseq at a series of time points across the optimised differentiation protocol. Subsequently, the hiPSC-TEP produced in the optimised protocol were tested in two functional assays – a 2D co-culture system, and a 3D ‘organoid’ model. Overall, the approach taken was interesting and based on sound rationale (although this appeared not to be applied consistently in the choice of factors), but the outcome was rather disappointing. While epithelial cells that expressed some markers characteristic of thymic epithelial cells were clearly produced it is not clear from the data presented how well (or whether) these cells could support T cell development (since the required controls are not shown), nor how well they represented the epithelial subsets present in the human thymus (indeed while the data show that mTEC-like cells may have been generated, they do not provide convincing evidence for the generation of mature cTEC). This diminishes enthusiasm for the manuscript. Therefore, although credit is given for the rational approach taken and the authors are applauded for presenting a more in depth analysis than many of the papers recently published in this area, it is not clear that the current work represents a major step forward over the best currently published protocols (e.g. Ramos et al 2023 PMID :36963390).

Specific comments are provided below:

General

- Throughout, the clarity of the manuscript could be improved, it is is rather confusingly written in several places. Typos should also be tidied up including in the figures. [1]
- Please show negative controls for all immunofluorescence staining and also for the flow

cytometry analyses presented, along with the full gating strategy for the latter (these can be provided as supplementary figures). [2]

- Please make sure that n is stated for all analyses, including immunofluorescence. [3]

[1] We thank the reviewer for raising this point, we made sure to clarify the main text and the figure legends that were significantly revamped.

[2] IF negative controls were included, with full staining on control cell lines, with particular focus on KRT5, PAX9 and FOXN1 staining, as questioned by the reviewers.

Due to the low quantity of available material at the late organoid stage, we used unstained controls for flow cytometry. We provide FACS plots of unstained controls and full gating strategies in supplementary figures.

[3] The number of biological replicates has been added to the figure legends, notably for qPCR and immunofluorescence experiments.

Specific comments:

Fig 1: The approach taken is appropriate (and welcome) and appears to have been applied well. Overall, the differentiation protocol shares similarities with previously published protocols. There are some key differences though (e.g. activation or inhibition of BMP signalling in the DE-3PP and 3PP-TEP stages, etc) and discussion of these differences would be appreciated [4], as would a summary of the precise optimised differentiation conditions (in the text and in Fig1e) [5]. A weakness is that, although the aim was to be guided by the outcome of the combinatorial analysis, some factors appeared to be added to the 'optimal' medium for each set because they were already included in established protocols, even when the current analysis suggested no benefit [6].

[4] Discussion has been amended with the following paragraph to discuss the main differences between our protocol and the two cited studies from Ramos et al. and Gras-Pena et al., mainly about SHH and BMP signaling modulation (**Line 708**):

"Recently, several groups reported generation of human TEPs from iPSc (Ramos et al., Gras-Pena et al.). By testing the effect of factor combinations on marker gene expression, they observed similar effects of BMP4 and RA supplementation compared to our study. Interestingly, Gras-Pena et al. showed that BMP signaling is time dependent and that a switch from inhibition at D15 to activation at D21 is beneficial for TEP induction. This is reflected in our protocol by the increase of the BMP4 dose between the 3PPE and TEP stages. Our findings are also consistent with the protocol proposed by Ramos et al. regarding the WNT-

BMP-FGF signaling axis at the TEP stage. However, both the Ramos and Gras-Pena studies highlight the importance of the SHH axis during TEP differentiation. We investigated the effect of SHH inhibition using cyclopamine at the AFE stage with no significant results. However, given the similar observations reported by Ramos and Gras-Pena, our protocol would benefit from a systematic study of precisely timed SHH modulation, using more potent factors than cyclopamine, such as SAG (SHH agonist) and SANT-1 (SHH inhibitor), to improve TEP differentiation.”

[5] Detailed optimized differentiation factor concentrations and timings are summed up in **Fig 1f**, with additional details about the conditions added to the main text: “Results of the DOE optimization are synthesized in...” (**Line 240**).

[6] DoE is a powerful methodology for simultaneously delineating the effects of multiple experimental factors. Based on this property, we argue for a more systematic use of statistics-based tools for designing iPSc differentiation protocols. However, one must remain aware of the limitations of this approach, particularly when using low-power designs such as Plackett-Burman, which are incomplete experimental plans focused on testing multiple factors at minimal cost to identify the most important ones. In addition, while the original project was to base the entire differentiation protocol optimization on DoE, we made interesting empirical observations regarding the effect of some factors on cell growth that were unexpected and could not be captured by our experimental readout. Indeed, our readout compared the transcriptome of the differentiation product to human pharyngeal development cell atlases, assessing the “quality” of the cells while entirely omitting the quantitative dimension. **New Supp Fig 4b** shows the significant effect of EGF and FGF10 supplementation on cell growth, justifying keeping these two factors in our differentiation protocol.

Another limitation of the DoE approach we followed concerns the readout. By capturing transcriptome profiles at each step of differentiation, we hypothesized that achieving a purer cell composition at each of these steps would systematically result in a higher generation of TEPs. However, one could imagine a scenario in which two differentiation products with similar purity at the 3PPE stage yield different numbers of TEPs. Effects like synergy by secretion of differentiation or survival factors by other cell types in the culture can explain such phenomenon. Thus, we tested whether the combination of BMP4 and FGF8 at the AFE to 3PPE transition - reported to have beneficial effects in the literature despite showing a neutral effect in our DoE - would enhance TEP generation by quantitating D15 *FOXN1* expression by qPCR as a direct assessment of the differentiation efficiency rather than an intermediate proxy. Interestingly, we observed a significantly higher expression of *FOXN1* in TEP samples

exposed to FGF8 and BMP4 at the earlier AFE stage (**new Supp Fig 4a**). Thus, the combination of BMP4 and FGF8 at this stage seems beneficial to later TEP differentiation, without directly affecting differentiation efficiency at the 3PPE stage, thereby justifying keeping these two factors in the final protocol.

We added this part to the main text: “We therefore specifically tested these two conditions by measuring D15 FOXP1 expression...” (**Line 213**).

Fig 2: The data presented in this figure assess how well the hPSC-TEP/TEC phenocopy primary human TEC. However, in most of the modules highlighted, there appears to be little correspondence with primary TEC (Fig 2) [7]. The fitting to the Magaletti data suggesting cTEC identity is interesting (Fig 2c), if surprising based on the heatmap data. However, the authors should provide details of the parameters used to allow superimposition of the two datasets in order that these data can be properly evaluated by the reader [8].

[7] The heatmap (**revised Fig 2a, Left**) displays the expression of differentially expressed (DE) genes in each iPSc-derived sample relative to D0 iPSc as a control. This analysis aims to characterize the differentiation itinerary by identifying top DE genes at each stage of differentiation and performing an unbiased kmeans clustering to group these genes into modules. Consequently, most gene modules are expected to be predominantly expressed at the differentiation step where they were identified as DE, rather than in the final differentiation product or primary TECs.

To better illustrate the differentiation itinerary, we generated a new heatmap using a new set of kmeans clustering parameters. We observe that the 7th module contains genes traditionally associated with immature TECs, even if the genes associated with some TEC functions, such as interaction with lymphocytes (**revised Supp Fig 6**), are located in the 8th module, which is expressed at lower levels than in primary TECs. Moreover, many genes in the TEC-expressed module 8 are markers of the T lineage with no expression in the iPSc, explaining the high normalized expression of this DE gene module. This figure highlights that, while we successfully reproduce the thymic differentiation path *in vitro*, the resulting TECs are immature.

[8] We fully acknowledge that representing deconvolution results, a vector of predicted proportions by cell population, on the UMAP projection of a scRNA-seq dataset can be confusing. Therefore, we have modified the figure to provide a more rigorous visualization, using a strip plot to display the predicted population proportions in the cell mix of the D14 differentiated TEP samples. This analysis and representation confirm that the main predicted

cell identity is TEC, which predominantly exhibits a cortical identity, with very limited contamination from immature esophagus, pharynx and thyroid lineages (**new Fig 2b**).

Figure 2a:

- Please provide more information as to how the modules were identified. [9]
- PSMB11, PRSS16, HLA-DR, CD80, CD86 are mentioned in the text but not annotated on the figure, please correct this omission. [10]

[9] Sample transcriptomes at each stage of differentiation were compared to D0 iPSc using DEseq2. Significantly DE genes with $\text{padj} < 0.01$ were merged in a list of significant genes. The expression of these significant genes was visualized using a heatmap of normalized expression across the samples of interest and compared to human primary TECs as positive controls. Genes were clustered into nine modules based on similar expression patterns using an unbiased kmeans clustering approach, allowing efficient visualization of gene sets upregulated at specific differentiation stages.

[10] *PSMB11*, *PRSS16*, *CD86*, *HLA-DRA* and *CD80* are not annotated in the Heatmap because none of them are significantly DE at the threshold of $\text{padj} < 0.01$ in any of the differentiation samples compared to D0 iPSc. This has been specified in the main text (**Line 262**). Nonetheless, we are not surprised that functional genes of mature cTECs, such as *PSMB11* and *PRSS16*, or antigen presentation-associated genes, such as *CD80*, *CD86*, and *HLA-DRA*, are not differentially expressed in the differentiated TEP samples, as no expression was expected in an immature TEC differentiation product. We later observe *CD86* and *HLA-DRA* expression in the TEC-like fraction of organoids. (**Fig 6d and new Fig 6g**).

- What does the un-annotated module (above 'EPCAM' module) represent? [11]

[11] In our revised heatmap, this module corresponds to modules 6 and 7 (**revised Fig 2a**). It contains DE genes that show preferential expression in our differentiation products at later stages (post D14), with lower expression in iPSc and primary TECs. These genes are enriched for GO terms associated with the extracellular matrix, collagen processing, and actin organization (**revised Supp Fig 6**), which is consistent with an epithelial, fibroblastic or even myoid identity. These clustered genes may be expressed by a fraction of improperly differentiated cell populations, as suggested by the deconvolution analysis at the TEP level (**new Fig 2b**). Otherwise, these genes would likely have clustered with the TEC associated genes (modules 8-9) or 3PPE genes (module 5).

- Why are the genes in the last module so over-represented? [12]

[12] Genes of the 9th module (module 8 in the modified heatmap clustering), which show the highest relative expression in primary TECs, are over-represented compared to the other samples. This is because they correspond to genes specific to mature TECs, with some appearing to be thymocyte specific, as indicated by the enriched GO terms in **revised Supp Fig. 6**. Possible interpretation is that (1) our differentiation product never expresses these genes at such levels, consistent with an immature differentiation state, or (2) a T cell contamination, potentially due to engulfed T cells in some primary TECs, leads to the presence of lymphoid-specific transcripts, whose genes are not expressed in our epithelial-lineage differentiation.

- These data do not convince that the protocol has developed TEC, although the lack of some canonical TEC markers in the primary TEC dataset suggests there may be a problem with those data. The hiPSC differentiation data could be checked against human TEC data in the public domain (e.g. from the Teichmann lab). [13]

[13] To assess the quality of the human primary TEC samples used as control for the iPSc to TEP differentiation, we checked whether these cells express canonical markers of the TEC cluster from the Park et al.'s human thymic cell atlas. This was done using a MA plot of DE genes in primary TECs vs iPSc (**new Supp Fig 5b**). We observed that classical TEC markers are significantly more expressed in primary TECs compared to all other samples, at the padj threshold of 0.05. The FACS gating strategy used to sort these samples is shown in **new Supp Fig 5a**, with an EPCAM / CDR2 / HLA-DRA staining. CDR2 was used by the Kyewski lab to identify human cTECs. As expected, we observed cTEC, mTEC^{lo} and mTEC^{hi} populations. Given the rarity of the TECs and the use of the low-sensitivity DGE-seq technology, we sorted the EPCAM⁺ TEC subpopulations together to ensure we obtained enough cells for reliable transcriptomic analysis. While this does not prove that our sorted primary TEC control samples are 100% pure or entirely free from contamination by other cell types (e.g. associated thymocytes inside thymic nurse cells), we consider these data robust enough to be used as control in our differentiation process.

Fig 2c:

– please provide details of how the deconvolution analysis was done i.e. what parameters were used to fit the bulk RNAseq data onto the Magaletta data? If I understand correctly, the Magaletta data are scRNAseq, whereas the dataset generated herein (for Fig 2) is bulk RNAseq. No details appear to be provided as to how these datasets were merged, again making it hard for the reader to evaluate their significance. [14]

[14] Here, we used bulkRNA-seq deconvolution with MuSiC to predict cell proportions in our bulk samples, using a scRNA-seq reference for comparison. Because visualizing the results with a UMAP is not suitable, we modified the figure to display a strip plot of the prediction results (**new Fig 2b**).

A sentence has been added to the main text to clarify the method used: “To predict cell type proportions in the bulk samples, deconvolution was performed using multiple scRNA-seq atlases as references (from Magaletta et al., Bautista et al., Park et al.), applying the MuSiC `music_prop()` function with default parameters.” (**Line 956**).

Related, SF5: The staining with FOXP1 looks real, the PAX9 stain is OK though less convincing, however, I find it surprising that all cells stain with these markers. Please provide negative control staining for both antibodies. [15]

[15] IF negative controls were included for the required markers using cell lines that do not express *FOXP1* (HEK293, hTERT/RPTEC) and *PAX9* (hTERT/RPTEC) in **revised Supp Fig 7a, Bottom**. Please note that we initially used HEK293 as a negative control for *PAX9* as well, but observed positive staining, which is consistent with *PAX9* exhibiting some expression in HEK293, as reported in the Human Protein Atlas database.

Figure 3

Fig 3a - please clarify whether the controls are unstained, or FMOs. [16]

[16] These are unstained controls. **Fig 3a** has been modified to include this clarification.

Fig 3b - the CD205+ TEP do not appear to resemble TEP or TEC very closely, but appear to strongly express a signature associated with mTEC fate. Please show the expression patterns of cTEC associated genes (especially since CD205 is cTEC affiliated later in development), and comment on these in the text. [17]

Also relevant here is line 296 – the Krt19+ population at E12.5 is a separate mTEC-fated progenitor, not a bipotent TEP, as shown by Lucas et al and also by Farley et al 2023, and evident in Magaletta etc, while the text in line 296 rather suggests that it is part of a common TEPC signature. [18]

[17] cTEC marker genes were identified from the Magaletta study, and their expression in the analyzed samples (end-stage differentiated TEPs and CD205+ TEPs) was visualized using a comparative scatter dot plot in **new Fig 3d**. All cTEC marker genes, except *IL7*, show lower expression in the sorted CD205+ TEPs. Combining this result with the deconvolution

prediction, which is enriched toward the mTEC lineage (**new Fig 3c**), supports a medullary fate for the CD205⁺ TEP in addition to the cortical one. This has been specified in the main text : “Thus, the CD205⁺ TEP population seems to be preferentially...” (**Line 332**).

[18] Main text has been modified to reflect reviewer’s suggestions: “TEPs exhibit significant cTEC/mTEC lineage plasticity...” (**Line 308**).

Fig 4: Here, the authors set out to push the PSC-‘TEC’ towards the mTEC fate using RANKL, as well as IL7, SCF and FLT3L (Of note, it is well established that in vivo RANKL does not regulate the cTEC/mTEC fate choice but allows mTEPC to expand). [19]

[19]When designing the plans and experiments in late 2019, we decided to supplement the organoids with RANKL, based on the prevailing consensus that RANKL was necessary for mTEC fate commitment (Pinto et al.). We added IL7, FLT3L and SCF due to their well-established roles in supporting early thymocyte survival and expansion, to complement the endogenous secretion by differentiating TECs. Recent advances indicate that RANKL drives expansion of the already committed mTEP population (Farley et al., 2023). This effect is compatible with a beneficial role of RANKL in increasing medullary cell growth, as highlighted by the deconvolution of CD205⁺ TEPs. This analysis shows that nearly one third of the cells with inferred TEC signatures are medullary oriented (~30% cTEC, ~15% mTEC) (**new Fig 3c**). We modified the text to reflect this nuance: “We selected these factors due to their well-established role...” (**Line 373**).

- The ‘ETP’ population used for the co-culture is Lin-CD34⁺CD7⁺ (lin= CD3 CD4 CD8 CD14 CD19 CD56), and, from the analysis shown in supplementary figure 6 a and b appears to contain some cells (~10%) that are starting to express CD4, and some cells that express CD3. Based on this, it seems more appropriate to call the sorted population ‘DN’ rather than ETP – although the majority of the CD4⁻ fraction of sorted cells appear to be DN1, there are clearly other cells in the sorted population including cells that may have passed the beta selection checkpoint. Please clarify this in the text. [20]

[20] To address the reviewer’s remark regarding the lack of specificity in the primary ETP sorting, which challenges the claim of full T differentiation in our model, we included a new backstaining experiment in **new Supp Fig 9a**. This experiment was performed on the ETP population sorted using the same strategy described in the paper and illustrated with four additional examples in **new Supp Fig 8**. We observe that no CD4⁺ or CD3⁺ cells have contaminated the population of interest, ensuring that the sorted ETPs do not contain

thymocytes that have matured beyond the DN1 stage, and have not passed the T lineage commitment and β selection checkpoints, nor undergone rearrangement of the TCR's α , β and γ chains. However, our ETP sorting strategy may yield varying purity levels, likely due to the heterogeneity of early thymocyte populations in the donor and experimental variability. This has been added to the main text (**Line 360**).

As pointed out by the reviewer, the phenotype of the sorted cells CD45⁺CD3⁻CD4⁻CD8⁻CD34⁺CD7⁺ is consistent with a heterogeneous mix of early thymocyte populations, including TSP2 cells arriving in the thymus, ETPs, and their immediate progeny at the double negative (DN)1 stage. For the sake of clarity and rigor, we will define this mixed population of thymocyte progenitors as ETP in its broader sense throughout the manuscript.

This has been specified in the main text (**Line 125**).

The authors observe association of the 'ETP' with TEC, and expression of K14, K5, MHCII and Cldn3. They state that this phenotype represents a fraction of the differentiated cells, but from the data presented it looks like most of the differentiated cells express these markers. Please clarify the proportion of cells in each dish/on each slide which are positive for the above markers. [21]

[21] We used larger field IF images of the same differentiation experiment to quantify differentiated cell populations based on CLDN3 expression (**new Supp Fig 10b**). CLDN3 expression has been shown by Park et al. in their human thymic cell atlas to be restricted to the most mature medullary compartments (mTECII–mTECIII) (**new Supp Fig 10a**). *KRT14* and *KRT5* exhibit a broader pattern of mTEC expression (**new Supp Fig 10a**). We quantified CLDN3-positive cells in a larger field IF and identified 14% of the total cell population, indicating heterogeneity in the differentiated population and relatively low maturation of TEPs in 2D. This result aligns with the low number of HLA-DR positive cells, which were extremely sparse in IF and were quantified by FACS analysis for more precise results (**new Supp Fig 10c**). This analysis supports the low level of mTEC maturation occurring in 2D co-cultures.

This has been specified in the main text: "We counted cells positive for CLDN3 in a larger field..." (**Line 395**).

Fig 4 does not examine what happened to the 'ETP' in the co-culture, did they differentiate? Please comment on this in the text and show related data. [22]

[22] We replicated the TEP + ETP 2D co-culture experiment. After three weeks of culture, the cells were passaged and stained with the thymocyte maturation panel to evaluate T cell

differentiation (**new Supp Fig 11**). We observed fewer CD45⁺ ETP-derived cells in the 2D co-culture than expected, potentially due to cell death or insufficient stimulation during culture. In addition, we detected minimal differentiation into CD3⁺ cells, most of which were CD4⁺CD8⁺ cells. This suggests that while the 2D co-culture can support ETP maturation to some extent, its efficiency is an order of magnitude lower than that of the 3D organoid system.

We added this result in the main text (**Line 407**).

Figure 5

- It seems somewhat surprising that the authors choose to use a set of markers of 'TEP' (CDH10, ACKR3, TBX3 and TGFB2) that does not include any of the markers normally taken to characterise the TEP populations present in early development (or indeed the various TEP populations suggested in the adult thymus). Inclusion of some commonly agreed TEP markers in this analysis would be welcome. [23]

[23] We originally selected four markers specific to iPSc-derived TEPs by comparing their transcriptome to that of the iPSc they derived from. We aimed to narrow the original set of marker genes to those that could specifically distinguish generated TEPs from natural TEPs based on their expression, thereby explaining the selection of the uncommon gene signature shown in previous Fig 5d. We selected two surface membrane protein coding, one signaling factor and one transcription factor genes that were significantly upregulated in the generated TEPs. According to the reviewer's suggestion, **Fig 5d** has been revised to feature a more classical TEP signature using common markers from Magaletta et al. and Farley et al. 2023: *PAX9, PDPN, FN1, EBF1, IL7*. This new set of markers hasn't altered the interpretation of the figure.

- It is not clear that the Monocle analysis adds much useful information, the authors could consider moving these data to a supplementary figure. [24]

[24] According to the reviewer's suggestion, the figure was moved to **new Supp Fig 12b** and the related text shortened (**Line 440**).

It is not clear whether in Fig 5f, the cluster 1 and 2 cells are part of a population that is present in normal human TEC, or whether they (and indeed the cluster 3 cells) are clustering separately but adjacent to the ex vivo TEC populations. It appears to be the latter, as all/almost all of the cells in those areas represent the hIPS-'TEC'. Please clarify this in the text, not least as it affects the strength of the conclusions that can be drawn. [25]

The data shown in Fig 5f (from d7 organoids) do not strongly support the notion that bona fide TEC have been made in this protocol (rather, they suggest that cells that have some resemblance to immature cTEC have been made, but that these have not generated normal differentiated TEC populations. A few cells resembling but not identical to mTEChi cells are also present). Please comment. [26]

[25 - 26] The scRNA-seq analysis of D7 organoids shows that the majority of cells belong to an immature cTEC-like population (cluster 1 and 2), exhibiting transcriptomic similarities with cTEC from the human thymic stromal atlas used as reference. A minor population (cluster 3) shows transcriptomic similarity to the AIRE⁺ mTEC population. While this is not direct evidence of mTEC^{hi} differentiation, it suggests that these cells share, at least partially, the transcriptomic program of mTEC^{hi}. Consistent with mTEC^{hi} features, cluster 3 cells exhibit a potential for TRA expression. The difference between cluster 3 cells and AIRE⁺ mTEC may be attributed to suboptimal *in vitro* culture conditions, such as the lack of integration with other *in vitro* cell types, including the innate immune compartment or endothelial vasculature.

This has been specified in the main text : “This indicates that the induced TECs ...” (Line 454).

Fig 6

- See Line 398 – that statement that thymocytes were concentrated around the projections does not appear to be strongly supported by the data shown. Rather, thymocytes appear scattered through the space examined (see Fig 6b, lower left panel; these are d28 data). Please provide quantitative data confirming a positive association, or tone down this comment. [27]

[27] We performed colocalization quantification manually using FIJI. We observed that only a slight majority (mean 65%, n = 3) of thymocytes colocalized with the organoid projections, and that many of them freely circulate within the hydrogel without interacting with the epithelial compartment of the hTOs (Supp Fig 12c, Left), which is a limitation of the system. We thus tone down this part in the main text : “some degree of colocalization..” (Line 488).

- Line 402 – please bear in mind that KRT5 does not necessarily indicate mature epithelium. There are plenty of examples (including in fetal thymus) where it marks stem or progenitor cells. The text should be adjusted accordingly. [28]

[28] The reference to KRT5 as a marker of mature TECs has been removed and the main text adjusted according to the reviewer’s suggestion (Line 387). Indeed, the human thymic cell atlas from Park et al. does not show KRT5 expression to be restricted to the more mature TECs (mTECII - mTECIII), but rather highlights it as a general mTEC marker (new Supp Fig

10a). To confirm that KRT5 emerges during TEP-to-TEC differentiation, we performed immunofluorescence experiments showing that TEPs are negative for KRT5 (**new Supp Fig 12d**).

- In Fig 6e, without knowing more details of how the deconvolution was done, it is hard to know how significant these data are. Please provide this information. It looks as though the hiPS-‘TEC’ share features with cTEChi, and with some minor mTEC populations, but not with the major mTEClo and mTEChi populations. [29]

[29] We applied the same methodology as for the deconvolutions shown in **new Fig 2b and 3c**, using MuSiC to deconvolve the HLA-DRA^{hi} and HLA-DRA^{lo} iPSc-derived TEC subsets of hTOs onto the human TEC atlas from Bautista, which integrates the Park TEC dataset (**Fig 6e**). As controls, we added RNA-seq data from sorted human mTEC^{hi}, mTEC^{lo} and cTEC populations obtained in collaboration with Part Peterson’s lab. These controls mapped correctly back to the scRNA-seq reference annotation, validating the approach. Indeed, the deconvolution predicts balanced proportions of cortical (mainly mature) and medullary TECs, the latter comprising multiple subpopulations.

A weakness of the bulk RNAseq approach is that it is not possible to determine the phenotype of single cells or the proportional representation of ‘TEC-like’ cells within the organoid. In view of this (see also comments above), the conclusion drawn (line 439) is overstated. Please rephrase to avoid overclaiming. [30]

[30] To avoid overstatement, the conclusion has been modified (**Line 543**) to: “Together, these findings showed that our differentiation product acquired a phenotype close to that of human TECs, with differentiation into distinct subpopulations exhibiting cortical- and medullary-like features, as well as the capacity to present self-antigen peptides to developing thymocytes through high MHCII expression.”

Please show the negative controls for the flow cytometry stains, as a supplementary figure (preferably FMOs) [31]

[31] We added unstained controls for the flow cytometry experiments due to the limited amount of material available for FMO acquisition. We notably provide CD4 and CD8 negative controls (**revised Fig 7d, Bottom; new Supp Fig 9a, 11, 14c, 17, 21**), as well as CD205, EPCAM and HLA-DR/DP negative controls (**new Supp Fig 10c, 13**). For the new iPSc-derived hematopoietic stem and progenitor cells (HSPC) experiments, FMO controls were performed

for phenotypic comparison (**new Supp Fig 20b**) along with unstained controls (**new Supp Fig 20c,d**).

Panels a and c – the levels of CD3 look consistent between weeks 3 and 5 in panel a (and fairly low), but are much higher in week 3 in panel 5. Please comment. [32]

[32] We acknowledge significant inter experimental variability due to multiple factors, such as the complexity of the protocol, the inherent variability of directed iPSc differentiation, and the use of primary hematopoietic cells from different donors. This variability is visible in **Fig 7b** and most pronounced at day 21.

- Please show CD4 v 8 profiles at timepoints earlier than week 5 (as FACS plots, not just a bar chart). Please comment on the lack of double positives at week 7 (panels d and e) - to my eyes, there are none in the plot shown in panel d or the upper plot in panel e, and very few in the lower panel e plot. I realise these are gated on CD3 but nonetheless I would expect to see DP in this population. The plots shown appear to contradict the bar chart, in which DP is the predominant population at week 4 and week 5. In line 456, the authors state that most CD3+ cells are DN, but that 'CD4+CD8+ DP, CD4+CD8- SP4 and CD4-CD8+ SP8, albeit in lesser proportions (Fig. 7d)'. However, no DP are present in Fig 7d. This statement also appears to contradict the data shown in Fig 7f. [33]

[33] We generated several new TEP/ETP organoid experiments and observed consistent results with those presented as FACS plots in previous Fig 7d, e. These results show a significant CD3⁺ DN population, SP8 and SP4 cells at varying proportions and few DP cells, contrasting with the experiment represented in previous Fig 7f, where DN cells were low and DP cells were high. Upon reanalyzing this data, we identified an unusual massive mortality in the thymocyte fraction at week 4 and 5, which affected the observed thymocyte proportions. This sudden significant mortality may be due to the donor ETP genetics or unexpected deleterious conditions in the gel. We replaced the bar plot with detailed FACS plots from both previous and new organoid cultures. In addition, we included panels showing the average generation of thymocyte fractions over time (**new Fig 8**).

If the input CD45+ population had been pure ETP, this would not be a concern, but since 10% of cells in that population are starting to express CD4 (and some of which express CD3), I am concerned that the SP populations present may reflect in vitro maturation of already committed DNs (including some that may already have passed beta selection) driven by the cytokines provided. Please show representative CD4 v CD8 plots at each timepoint after gating on CD45

only, and also show the negative control data (i.e. culture of the DN in the absence of the hPSC-TEP, in the same medium as the organoids, related to SF7b). [34]

[34] We conducted multiple rounds of control experiments to address this crucial point. First, we observed DC differentiation in the hTOs, demonstrating the presence of uncommitted progenitors within the sorted ETP population. However, this does not exclude the presence of a minor already committed population that would take over during the differentiation and generate most of the T cells in the hTOs. To investigate this, we cultured the sorted ETP population in the same culture medium as hTOs but without thymic epithelial cells (**new Supp Fig 14c, Top**). This resulted in high cell mortality, with only 0.6% living cells, indicating a lack of survival signals. Furthermore, only an insignificant number of differentiated CD3⁺ cells were detected, demonstrating that co-culture with TEPs is necessary for ETP survival. We added this result in the main text (**Line 579**).

We then isolated the already committed DN2/3 thymocytes (CD3⁺CD4⁺CD8⁺CD34⁺CD7⁺) (**new Supp Fig 8**) and cultured them in the same medium in the absence of TEPs (**new Supp Fig 14c, Middle**). Cell survival is higher than with ETPs (11.3% living cells) but remained significantly lower than what we typically observed in hTOs. Among these cells, differentiation into CD3⁺ SP8 T cells was observed, indicating preexisting T-lineage commitment. However, they did not exhibit a mature CCR7⁺CD62L⁺ phenotype. Hence, the lack of similar differentiating thymocyte populations from ETPs cultured alone confirms the absence of committed DN thymocytes in the isolated ETPs and underscores the reliability of our ETP isolation strategy. This has been specified in the main text (**Line 585**).

Finally, as a final control to assess the ability of hTOs to drive T cell commitment and maturation, we generated hTOs by aggregating Hematopoietic Progenitor Stem Cells (HPSCs) derived from the same iPSc line as the TEPs (**new Supp Fig 19a**). The generated CD34⁺CD7⁺ HPSCs fully differentiated *in vitro* and did not receive any commitment signal. We performed the experiment in duplicates, and observed high cell viability (~95% at D28, and ~74% at D35), as well as differentiation into CD3⁺ cells (~28% at D35), which were almost entirely composed of TCRαβ⁺CD8⁺ T cells (**new Supp Fig 21**). We also observed myeloid lineage differentiation at levels comparable to those in hTOs (**new Supp Fig 22**). We added this result in the main text (**Line 657**).

Together, these new results indicate that the ability to drive T cell differentiation is inherent to the hTO system and is neither preexistent in the initial progenitor population nor induced by culture medium supplementation.

As requested by the reviewer, we included CD4 vs CD8 plots from both previous and new hTO experiments at several time points, after gating on CD45⁺ alone and on CD45⁺CD3⁺ for comparison (**new Fig 8a**). The analysis revealed an overrepresentation of CD4⁺ and DP cells in the CD45⁺ fraction compared to the CD45⁺CD3⁺ fraction, which was pronounced at D27 and gradually decreased over time, reaching equality at D38. This suggests that ISP cells (CD4⁺) and their immediate DP progeny are captured in the CD45⁺ fraction and mature within the hTOs, acquiring CD3 and progressively reaching SP4 and DP cell numbers comparable to those in the CD3⁺ fraction. We added this result in the main text: “we observed an overrepresentation of CD4⁺ and DP cells...” (**Line 595**).

Line 451 – can the authors really say commitment here, as they really start with DN rather than ETP (some of the ‘ETP’ will already have undergone commitment). Please modify the text as needed. Related, please show thymocyte subset data for the controls (SF7b) rather than just cell numbers; these data are essential. [35]

[35] Flow cytometry data of experimental controls for the effect of TEPs on ETP-to-thymocyte differentiation are shown in **new Supp Fig 14c, Top**. We confirmed significant ETP and ETP-derived cell mortality, as well as the absence of a CD3⁺ population. These observations indicate that the culture medium including its cytokine and growth factor supplementation is not sufficient to provide the necessary survival and growth signals. Moreover, this further supports the purity of the sorted ETP population. Indeed, the same control experiment with DN cells showed higher thymocyte viability and the presence of a CD3⁺ SP8 population, though lacking CD62L and CCR7 expression, markers of T cell maturation. The FACS plots for the controls are shown in **Supp Fig 14c, Bottom**.

Minor comments:

Line 74 – The literature on TEPC has moved on a bit from these two references, please update. In particular, whether there is a common TEPC that plays a major role in generating or replenishing TEC remains rather open. [36]

[36] The main text has been reworked to reflect the current state of the literature with the addition of three references (Nusser et al., Bleul et al., Ulyanchenko et al.):

“Although the mechanisms supporting the differentiation of mTECs and cTECs remain elusive, recent studies have identified one or several bipotent thymic epithelial progenitor cell (TEP) populations capable of giving rise to complete and functional cortical and medullary compartments (Nusser et al., Bleul et al., Ulyanchenko et al.). Studies have identified a

bipotent TEP population exhibiting a cortical-like phenotype (Alves et al., Baik et al.). NOTCH (Li et al. Liu et al.) and RANK-CD40-LTB (Irla et al. Lopes et al.) signaling appear to play a crucial role in regulating TEC fate decisions.” This has been added to the main text (**Line 71**).

Line 76 – The Notch signals described in these papers include Notch signalling that is active pre-haematopoietic colonisation, so can't be provided by cross-talk. Please clarify the text accordingly. [37]

[37] Indeed referring to thymic crosstalk at this stage is incorrect due to a mismatch of the timing of hematopoietic colonization. This part was removed from the main text (**Line 79**).

Line 90 – Please make sure the referencing for the directed differentiation approaches is complete. [38]

[38] We have added the following reference: Ramos et al. 2023.

Line 269 – its not clear at what point, if ever, in early development there are bipotent progenitors, though both cTEC and mTEC progenitors express b5t (and presumably, CD205). See Magaletta et al 2022, Farley et al 2023, Liu et al 2020, Li et al 2020, Nusser et al 2022 among others. Please correct this statement to accurately reflect the literature. [39]

[39] The main text has been modified and references updated as suggested. The paragraph (**Line 308**) is now introduced with the sentence: “TEPs exhibit significant cTEC/mTEC lineage plasticity and studies have shown that these progenitors express CD205”.

Line 313 – the authors say ‘in addition to a cortical one’ [fate] but to my mind, differentiation into a cTEC fate has not been adequately demonstrated, by the authors' admission the cells generated lack canonical cTEC markers. Please correct accordingly. [40]

[40] The main text has been modified (**Line 368**) to soften the claim regarding true lineage differentiation:

“To test whether the TEPs we generated have the potential to phenocopy TECs from the medullary lineage in addition to the cortical lineage...”

Line 330 – The data presented show that something like an mTEC may have been generated, though no details of which subtype(s) of mTEC are provided. There does not appear to be any convincing evidence that mature cTEC are generated. Please modify the statement on line 330 to reflect this. [41]

[41] The main text has been rephrased (Line 402) with “TEC-like cells, with a small fraction expressing markers associated with maturation (HLA-DR) and medullary fate (CLDN3)”.

Supplementary Figure 1d: please state in legend what 1 and -1 mean, I assume it means hi and lo dose but this is not clear – were factors also omitted? [42]

[42] The legend of **Supp Fig 1d** was modified as recommended: “The binary levels reflect doses (-1: low dose, 1: high dose) as described in c.”

No factor was omitted, the table is consistent with **Supp Fig 1c**, **Fig 1b** and **Fig 1d**.

Reviewer #2 (Remarks to the Author):

I co-reviewed this manuscript as part of the early career programme at nature communications.

#####

Reviewer #3 (Remarks to the Author):

The creation of in vitro systems that fully recapitulate thymic stromal features/function is essential for regenerating the thymus in pathological settings and in aging. However, capturing and maintaining the complexity of the thymic microenvironment in vitro from iPSCs, especially the various types of thymic epithelial cells (TECs), has been a challenge despite numerous attempts by several groups. Moreover, while recent studies reported in vitro systems capable of fully reproducing thymocyte development, some specific steps of thymocyte selection/maturation remain an obstacle, raising the need for a fine-tuning of current thymic organoid models.

In this report, Provin and colleagues take on the challenge and use an original methodology to rationalize the culture conditions required for differentiation of iPSCs to TEPs, coupling statistical-mathematical approaches to previously published gene expression datasets on early thymic organogenesis, to recreate optimal experimental designs. Focusing on 4 key steps of thymic organogenesis the authors were able to confirm/demonstrate which specific pathways previously used in the field were needed for the induction of TEP fates from iPSC. The authors also identified a new pathway (IGF-1) to be targeted in vitro for TEP induction. In

addition, they showed the higher efficiency of their protocol for TEP generation in comparison to two previously published reference methods, and its reproducibility across iPSC lines.

After extensive phenotypic and transcriptomic characterization of iPSC-derived TEPs, they proceeded to test their potential to further differentiate into TEC subsets in vitro, by aggregating cells with CD34+CD7+ human thymic progenitors to create human thymic organoids (hTOs). They briefly assessed the phenotype of TECs in later stage hTOs and show the possible presence of mature mTEC subsets. This maturation process was obtained in vitro, bypassing the need for in vivo transplantation of organoids usually described in the field. Finally, the authors identified CD8SP and CD4SP and dendritic cell subsets derived in hTOs.

This work is original in its DOE approach although this analysis relies on previously published protocols. While the methodology for deriving TEPs and hTOs has some novelty, the evidence showing this system can generate sufficient and functional T cells in a way that surpasses current models in the field is limited.

Below is a detailed list of major and minor comments/points to be addressed:

Major comments:

- There is a lack of quantitative data throughout the manuscript: how many TEPs can be obtained from iPSCs? [43]

How many cells are needed (TEPs and ETPs) to make hTOs? [44]

How many hematopoietic cells (T, DC) can be harvested from hTOs (the FACS profiles suggest this is very limited)? [45]

How reproducible is the hTO system in showing T cell output from TEPs? [46]

[43] **New Supp Fig 7b** has been added to illustrate the growth of the iPSc-derived TEPs using data from routine quantifications during the protocol. Cell culture expands from an initial density of 35,000 cells/cm² at D0 to a final concentration of ~600,000 cells/cm² at D13, at the TEP stage, equivalent to ~2.1 million cells per P12 well. This highlights the capacity of our protocol to generate TEPs in large quantities. This information has been added to the main text (**Line 288**).

[44] During the hTO reaggregation step in low-binding 96-well plates, we seeded 2,500 freshly sorted primary ETPs with 20,000 iPSc-derived TEPs. The main text has been modified to include this information (**Line 866**).

[45] We quantified the yield of the organoid system by FACS, harvesting cells from multiple organoids at several time points, and staining them with classical thymocyte markers (CD45, CD3, CD4 and CD8) to track the growth curves of each population. The generated plots have been added in **new Fig 8c**. In terms of order of magnitude, a single organoid seeded with 2,500 ETPs at D0 generated an average of ~15,000 CD45⁺CD3⁺ T cells at D40, including ~500 SP4 (CD45⁺CD3⁺CD8⁻CD4⁺) and ~4,000 SP8 (CD45⁺CD3⁺CD8⁺CD4⁻). Although the absolute yield of mature CD4 and CD8⁺ T cells per organoid remains limited, automating hTO formation and scaling up the system with large batches and optimized culture setups will enable robust and efficient T cell production. The main text has been modified to include this point (**Line 613**).

[46] We fully acknowledge the non-negligible variability of T cell proportion and yield between organoid batches, as well as between wells within the same experiment. **New Fig 8c** illustrates this variability in the T cell output. This variability can be attributed to the inherent complexity of the system, which relies on intricate interactions between cells differentiated from sensitive iPSc and primary cells from human donors, encapsulated in 3D hydrogels. Enhancing the standardization of organoid generation and replacing primary donor-derived hematopoietic compartment cells (ETPs) with iPSc-derived cells, could improve reproducibility.

- Clarification of what is different from other published protocols using the DOE revised methodology is essential to evaluate what is new in the paper. [47]

[47] The discussion has been amended with the following paragraph to highlight the main differences between our protocol and the two cited studies (Ramos et al. 2022, Gras-Pena et al.), particularly regarding SHH and BMP signaling modulation (**Line 708**):

“Recently, several groups reported generation of human TEPs from iPSc. By testing the effect of various factor combinations on marker gene expression, they observed similar results regarding BMP4 and RA supplementation compared to our study. Interestingly, Gras-Pena et al. showed that BMP signaling is time dependent and that switching from activation at D2 to inhibition at D15 benefits TEP induction. This is reflected in our protocol by an increase of the BMP4 dose between the 3PPE and TEP stages. Our findings are also consistent with the protocol proposed by Ramos et al. regarding the WNT-BMP-FGF signaling axis at the TEP stage. However, both Ramos and Gras-Pena studies note the importance of the SHH axis during TEP differentiation. We studied the effect of SHH inhibition using cyclopamine at the AFE stage and did not observe significant results. However, given the similar observations reported by Ramos et al. and Gras-Pena et al., our protocol would benefit from a systematic

study of precisely timed SHH modulation, using more potent factors than cyclopamine, such as SAG (SHH agonist) and SANT-1 (SHH inhibitor), to improve TEP differentiation.”

- Figure 7: the authors mention the presence of all thymocyte subsets in hTOs, but there is no data on the kinetics of T cell development from early to late time point in hTO cultures including the development of precursors, most importantly DPs. CD4 versus CD8 flow data would be an important addition to the bar graph in Figure 7f. [48]

[48] The development kinetics of different thymocyte populations was calculated from FACS data and is shown in **new Fig 8c**. We observe stable proportions of DN thymocytes throughout differentiation, with a slight decrease of SP4, and stable proportions of SP8 thymocytes. However, the number of generated SP8 and SP4 increases over time, indicating that their stable or reduced proportion among all CD3⁺ cells results from a greater expansion of the DN compartment resulting from ongoing ETP differentiation or enhanced DN proliferation. We performed several new TEP/ETP organoid experiments and observed results consistent with those presented as FACS plots in previous Fig 7d, e, showing a significant CD3⁺ DN population, SP8 and SP4 cells at varying proportions, and few DP cells, contrasting with the experiment we represented in bar plots with low DN and high DP in previous Fig 7f. We reanalyzed this data and identified an unusual massive mortality of the thymocyte fraction at weeks 4 and 5 that impacted the observed thymocyte proportions. This sudden and significant mortality may be due to the donor ETP genetics or unexpected deleterious conditions in the gel. We replaced this bar plot with the detailed FACS plots from previous and new organoid co-cultures we performed, including panels showing the average generation of thymocyte fractions over time (**new Fig 8a, b**).

This has been added to the main text (**Line 593**)

The authors describe the presence of mature T cells generated in the hTOs, based on the expression of CD3⁺ cells at selected time points but TCRab expression data is very weak. CD3 is also expressed by NKT cells so additional markers should be used to discriminate between T and NKT cells. [49]

What are the CD3⁺TCRa/b⁻ cells? Moreover, a better characterization of the maturation of SP4/SP8 cells in hTOs is needed (CD3 vs TCRab expression) at all stages of hTOs development. [50]

[49] To rule out differentiation into NKT cells, we quantified NK lineage marker expression in the scRNA-seq data of the non-adherent cells in our organoids (**new Fig 9e, Right**). Most genes exhibited absent or low expression in the three main clusters, except for markers that

overlap with T cells or DCs, such as *CD3E*, *CD44* or *CD69*. *PRF1* showed some level of expression, however not sufficient to indicate NKT differentiation, as it is also expressed to some extent in T cells (source: Human Protein Atlas). Thus, we can confidently rule out the presence of NKT cells in our system. The main text has been modified with “We ruled out the presence of NK...” (**Line 638**).

[50] Although some studies place the emergence of CD3 expression at the DP stage, with CD3⁺ and CD3⁻ DP cells (A Dik et al 2005 pmid: 15928199), the expression of signal-transducing CD3 is also described as part of the pre-TCR complex, composed of the β chain, CD3 and a pre- α chain, in stages as early as DN2 (Azzam et al., Boehmer et al.). This population could therefore represent maturing, pre-TCR $\alpha\beta$ DN T cells. This has been added to the main text (**Line 562**).

While the authors provide some phenotypic characterization of TEP, a more detailed phenotypic analysis of other stromal cells in 3D conditions (hTOs) throughout the length of the cultures is missing. In particular, the frequency of EpCAM⁺ and/or EpCAM⁺CD205⁺ cells between D0 and D35 in hTOs is not shown [51]

Is FOXP1 detected at early/mid/late stages of hTO cultures? [52]

[51] To address this question, we carried out hTO cultures and assessed the proportion of EPCAM⁺CD205⁺ cells at three time points: 7, 14 and 21 days after formation in the hydrogel. We also stained the cells for HLA-DR as a proxy of TEC maturation. As shown in **new Fig 6g** and **new Supp Fig 13**, the proportion of CD205⁺ and CD205⁻ cells within the EPCAM⁺ population is balanced, with a stable 45% of CD205⁺ cells at D7 and D14, before decreasing to 20% at D21. Importantly, we observe that a large majority of these CD205⁺ and CD205⁻ cells are positive for HLA-DR expression at D14 and even more markedly at D21, consistent with the enhanced HLA-DR and DP fluorescence at D21. We added these results in the main text: “To monitor the dynamics of EPCAM⁺CD205⁺ and HLA-DR⁺ cell populations during hTO differentiation...” (**Line 528**).

[52] *FOXP1* expression quantified by qPCR across TEP differentiation and hTOs is shown in **new Supp Fig 12a**. *FOXP1* is detectable from D11 and increases substantially in organoids. We added these results in the main text: “We first validated increased *FOXP1* expression...” (**Line 426**).

- Is there any spatial evidence of distinct cortical/medullary areas in hTOs? [53]

[53] We observed heterogeneity in D28 organoids with cellular regions composed of grouped DAPI positive cells that are not stained for CD45 or KRT5 (**Fig 6b** and **Supp Fig 12c**). Given the proportion of cellular populations analyzed by FACS in D28 organoids and the limited presence of EPCAM⁺CD45⁻ stromal cells (**Fig 6c** and **new Supp Fig 12e**), these unstained cells are at least partially consistent with an EPCAM⁺KRT5⁻ population, potentially representing cortical-oriented TEC-like cells. This suggests a degree of heterogeneity with regionalization in the hTOs. Investigating hTO spatial organization is within the scope of our ongoing research but requires overcoming limitations that currently hinder progress. To address this and achieve spatial characterization, we are working on adapting the Phenocycler (sequential IF) and Lightsheet imaging protocols to mitigate constraints imposed by hydrogel-induced autofluorescence and accommodate the unique filamentous organization of hTOs.

In an experiment aiming to favor TEP to TEC maturation and prolong ETP-mTEC interactions, we modulated our differentiation protocol by inhibiting BMP4 and adding activin A at the end of the AFE phase. BMP4 has been shown to promote the maintenance of TEPs in a progenitor state, while Activin A induces differentiation toward a medullary fate (Lepletier et al.). The interplay between these two cytokines has been reported to regulate TEP differentiation and maturation (Ramos et al. 2023, Lepletier et al.). Consistent with the literature, this experiment showed a differentiation bias towards a medullary phenotype. Throughout hTO culture, we observed a markedly reduced proportion of EPCAM⁺CD205⁺ cells compared to normal TEP-derived hTOs, supporting the otherwise cortical nature of these cells (**new Fig 6g** and **new Supp Fig 13**).

This point has been added to the main text: "hTOs were also formed with TEPs generated from a slightly modified differentiation protocol..." (**Line 535**).

Thymic cross-talk between epithelial cells and thymic progenitors/T cells has not been demonstrated and without stronger data this term should be deleted from the title and conclusions. [54]

[54] The title and the main text have been adjusted in several places to soften the claim regarding thymic crosstalk happening between epithelial cells and thymic precursors in our system (**Lines 79, 303, 685**).

- Quantification of TEP and hTO cell output (stromal and T cell) is essential to show the system is superior to others in the field. [55]

Stronger evidence of thymopoiesis including CD4+CD8+ development and kinetics of T cell differentiation is required as described above. [56]

[55 - 56] As detailed in point [45], we quantified organoid yield by FACS, staining for thymocyte markers (CD45, CD3, CD4, CD8) at various time points to track population growth (**new Fig 8c**). A single organoid seeded with 2,500 ETPs at D0 produced ~15,000 CD45⁺CD3⁺ T cells by D40, including ~500 SP4 and ~4,000 SP8. Automating hTO formation and scaling up the system with large batches and optimized culture setups will enable robust and efficient T cell production.

FACS plots displaying the thymocyte fraction of organoids stained for CD4 and CD8 have been added to **new Fig 8a** and **8b**. FACS plots showing the full gating strategy and the CD4/CD8 unstained controls have been included in **new Supp Fig 15-16** and **17**, respectively.

ETPs are a heterogeneous population, containing cells already primed for T-cell fate in the thymus. Of note, the cited paper Lavaert et al, 2020 defines the earliest thymic progenitors (TSP1) as CD7⁻ (by protein) and only CD7⁺ by RNA, so the authors have seeded TSP2 cells into the cultures, a population that has already committed to T cell lineage. Evidence that the TEPs produced from iPSCs in this system can support T cell development from less mature progenitor sources (cord blood CD34⁺ for instance) is required. [57]

[57] The phenotype of the sorted cells CD45⁺CD3⁻CD4⁻CD8⁻CD34⁺CD7⁺ is consistent with a heterogeneous mix of early thymocyte populations, including TSP2, ETP and DN1 - stages that have not yet passed the T lineage commitment checkpoint. Notably, the new restaining experiment (**new Supp Fig 9a**) confirmed that the so-called ETP-sorted population is also CD44⁺, a marker of early progenitor cells uncommitted to the T-cell lineage (Canté-Barrett et al. pmid: 28163708). This has been added to the main text (**Line 360**).

To support the claim that hTOs support T cell development even with less mature progenitors, we replicated the experiment using iPSc-derived Hematopoietic Stem and Progenitor Cells (HSPCs). Direct differentiation of iPSc into CD34⁺ HSPCs was conducted following the protocol established in our lab by Flippe et al.

At D9 and D12 of differentiation, we observed two distinct CD34⁺ populations: a CD34^{hi} compartment (mainly in the embryoid bodies), and a non-adherent CD34^{int} compartment resembling cord blood hematopoietic cells, predominantly found in the culture medium (**new Supp Fig 20a**). We noted that a subset of these CD34⁺ cells were also CD7⁺ (~40% in

embryoid bodies and ~60-90% in the culture medium) and CD44⁺ (~46% in the culture medium and 28% in embryoid bodies), resembling the phenotype of the primary ETPs used in our hTO system (**new Supp Fig 20b** and **c**).

At D12 of differentiation, we FACS sorted the CD34⁺CD7⁺ and CD34⁺CD7⁻ fractions (**new Supp Fig 20d**), harvested iPSc-derived TEPs from the same iPSc line, and formed CD7⁺ and CD7⁻ hTOs (as recapitulated in **new Supp Fig 19a**). After 24h, we observed the successful formation of spheroids in the CD7⁺ condition, and the absence of aggregation in the CD7⁻ condition. CD7⁺ hTOs were then seeded on hydrogels and cultured for 5 weeks, with FACS analyses performed at D28 and D35. The results, shown in **new Supp Fig 21**, revealed CD45⁺CD3⁺ SP4 and SP8 cells at D28, and SP8 cells at D35. This suggested that the SP4 population at D28 corresponds to the ISP stage that would entirely mature into SP8 mature T cells, predominantly TCR $\alpha\beta$ positive. We also observed that the CD45⁺CD3⁻ fraction contains CD14⁺ myeloid cells (**new Supp Fig 22**) in proportions similar to those found in primary ETP-based hTOs. Thus, this experiment, using iPSc-derived CD34⁺CD44⁺CD7⁺ HSPCs, further supports our model's ability to sustain the differentiation of immature hematopoietic progenitors into both T and myeloid lineages.

As per the reviewer's request, we repeated the experiment using purchased CD34⁺ cord blood (CB) cells. Upon hTO formation, the CD34⁺ CB cells successfully aggregated with TEPs and formed spheroids; however, the seeded hTOs failed to grow and acquire the expected morphology (**new Supp Fig 19b**). At D21, we observed spontaneous lysis of the CD34⁺ CB hTOs. A few living cells appeared to be CD45⁺ but CD3⁻. To better understand this outcome, we analyzed the CD34⁺ CB by FACS and found that 85% were CD7⁻ (**new Supp Fig 20b**). These new results were added in the main text (**Line 657**).

Both experiments point to CD7 expression being a key factor for successful aggregation and sustained differentiation of the hematopoietic compartment in our model. This is in accordance with the known role of CD7 in integrin-mediated adhesion of T cell precursors to thymic epithelial cells - potentially before CD3 and TCR expression (Shimizu et al. pmid: 1370688) - and in T cell proliferation (Shimizu et al. pmid: 1370688, Carrera et al. pmid: 2459196), as well as with the phenotype of *in vivo* TSP2 and ETP cells.

Flow cytometry data showing CD205 and HLA-DR expression in hTO TECs is unclear; a better characterization of the stromal compartment of hTO (and cellular compartmentalization) should be achieved by immunofluorescence [58]. Immunofluorescence experiments would better demonstrate the cross-talk between stromal and hematopoietic progenitor/cells instead of the current data shown in Figure 4. [59]

[58] As discussed in point [53], exploring hTO spatial organization is a key focus of our ongoing research but requires overcoming current limitations that hinder progress. To address this, we are optimizing Phenocycler (sequential IF) and Lightsheet imaging protocols to reduce hydrogel-induced autofluorescence and adapt to the unique filamentous structure of hTOs.

[59] As requested by the reviewer, we attempted immunofluorescence imaging on a 2D co-culture experiment to better illustrate crosstalk between the two compartments. On D7 and D10 of co-culture, small, round, and refringent T cells were visible and co-localized with TEPs under phase-contrast microscopy (**Fig 4b**). We carefully prepared the co-culture for immunofluorescence imaging (PFA fixation, Triton permeabilization, primary and secondary antibody incubation and DAPI staining) with minimal washing steps. However, by the end of the preparation, T cells were no longer visible under the microscope. Confocal imaging confirmed the absence of CD45⁺ cells. These findings suggest that the CD45⁺ compartment forms labile physical interactions with TEPs, which fixation disrupts through repeated medium changes, washing, and chemical treatments.

DLL1 and DLL4 expression are key markers of TECs. RNA-Seq expression of these are relatively weak in CD205⁺ TEPs whereas Jag1 and 2 expression appears to be more prominent (Fig 3b). Protein expression of these key markers (at least DLL4) should be given with appropriate controls. [60]

[60] To assess the expression of the crucial NOTCH ligands DLL1 and DLL4 at the protein level, we performed IF on D14 TEPs (**new Fig 3e**), using the HEK293 cell line as a negative control. We observed consistent DLL1 and DLL4 staining in differentiated TEPs, confirming NOTCH ligand expression and their potential to provide NOTCH signaling. This result also highlights the relatively low sensitivity of the DGE-seq, which enables 3' sequencing and multiplexing of 96 samples, compared to other RNA-seq strategies, such as the more expensive Illumina paired-end full transcript sequencing, even though both genes are detectable in these samples (**new Fig 3e, Right**). This has been added to the main text (**Line 338**).

The statement (line 258) that FOXN1 expression showed a complete differentiation into TEC-fate (Fig S5B) is misleading. Only a few cells are shown and no summarized data on the % of FOXN1 cells in culture is given. In addition, while FOXN1 is necessary for proof of fully differentiated TECs it is not sufficient. [61]

[61] We revised the statement (**Line 287**) to align with the reviewer's suggestion that the level of evidence provided is not sufficient to claim “complete TEC differentiation”. The sentence has been replaced with “indicative of differentiation toward TEC fate”.

FOXN1 IF staining on a larger field of view slide of the TEP culture, along with relevant negative control, is shown in **revised Supp Fig 7a**. Most cells are positive for FOXN1, with staining intensity varying among positive nuclei, possibly reflecting different differentiation stages within the population.

- Figures throughout need more complete explanation in legends. [62]

[62] We thank the reviewer for raising this point. The figure legends have been significantly revised and clarified, notably including essential details on the number of replicates and the statistical tests used.

Minor comments:

- Discussion: hTOs are said to contain fibroblasts, but there is no evidence showing the presence of fibroblasts in the Results section. Mesenchymal support of TECs would be expected to be required for maintenance of the hTOs. Where do the fibroblasts originate from? [63]

[63] While performing scRNA-seq analysis of the hematopoietic compartment in hTOs, we used a quick manual dissociation to collect non-adherent cells. However, we observed that a rare stromal population (cluster 1) was also included in the dataset (**Fig 9d**). SNP analysis confirmed that this population is iPSc-derived and expresses fibroblast markers *FN1*, *VIM* and *LUM*, clustering with the VSMC/fibroblastic cells of the human thymic stromal atlas (**Fig 9f**). This aligns with the presence of a CD45-EPCAM⁻ cell population in hTOs (**Fig 6c** and **new Supp Fig 12e**). We revised the text to be less conclusive about the identity of this potential fibroblast population. Indeed, these cells may arise as an unintended byproduct of TEP differentiation or result from TEC de-differentiation and phenotype loss. The discussion has been adjusted accordingly: “and a potential fibroblastic population...” (**Line 799**).

- Co-culture of ETPs was used to help mature the TEPs in vitro-discussion of how/if this would be used practically for generation of TECs for clinical use would be helpful. [64]

[64] Discussion has been amended (**Line 779**) with: “We foresee two main clinical applications for this system. The first leverages the differentiated TEC compartment to remediate rare pathologies where deficiencies in crucial genes involved in thymic organogenesis (such as

FOXP1 or PAX1 in SCID and *TBX1* in DiGeorge's syndrome) lead to thymic hypoplasia and immunodeficiency. Autologous transplantation with gene-corrected iPSc-derived TECs could correct this deficiency and restore, at least partially, T cell populations with self-tolerance. The second application leverages the ability of hTOs to mature T cells derived from iPSc banks for heterologous, off-the-shelf cell therapy, which also enables cell editing, such as CAR modification, for immuno-oncology applications.”

- Are there any regulatory T regs in hTO with CD4+ T cells? [65]

[65] We have included CD25 in our thymocyte maturation panel to detect a CD45⁺CD3⁺CD4⁺CD25⁺ Treg population. However, too few CD25⁺ cells were detectable to draw definitive conclusions about the presence or absence of Tregs, as natural Tregs represent a minor subset of the SP4 and SP8 compartments. Automating hTO formation and scaling up the system to increase the number of generated T cells will allow accurate Treg identification through direct FOXP3 detection and enable us to provide more conclusive evidence in the future.

- Inconsistent time point labeling: some figures specify D35, others W5. [66]

[66] Time point annotation has been revised and standardized throughout the manuscript to ensure consistency.

- Introduction: line 54-57 needs citations for the human TEC markers cited [67]

[67] Reference to Bautista et al. has been added to support this point (**Line 52**).

- Line 269: the statement that CD205 marks TEP that are bipotent should be clarified as CD205 is not specific - the ref for this should be cited [68]

[68] We modified the text to be less assertive about the relevance of CD205 as a marker of bipotent TEPs (**Line 308**): “TEPs exhibit significant cTEC/mTEC lineage plasticity, and studies have shown that these progenitors express CD205 (Magaletta et al., Farley et al.). Thus, we investigated whether the iPSc-derived TEPs we generated expressed the CD205 surface marker and in what proportion.

- Careful editing of grammar and meaning is needed throughout: the use of “remarkably” should be kept to a minimum if used at all. [69]

some examples in the Intro:

Line 77: Understanding of regulationhas shown progress

Line 86-understanding of regulation...has shown progress

Line 100-this is confusing as most forms of SCID are due to mutations in hematopoietic cells and so will not benefit from TEC replacement

Line 105-confusing example of application of TECs

[69] We thank the reviewer for pointing out our repeated use of the word “remarkably” in the main text. We have removed most instances of the word and carefully reviewed the grammar and overall writing throughout the main text and figure legends.

Regarding the specific requests:

Previous Lines 77/ 86 (**Line 90**): we replaced “recent progress” by “advances” to avoid repetition.

Previous Line 100 (**Line 104**): we revised the sentence to provide more specificity regarding the SCID subtypes.

Previous Line 105 (**Line 107**): we removed the suggestion to use hTOs for autoimmune pathologies such as APECED due to the speculative nature of the approach.

Reviewer #4 (Remarks to the Author):

#####

Reviewer #5 (Remarks to the Author):

In recent years, many techniques have been reported to induce differentiation and functional maturation of TECs. However, the myriad of protocols has left a need for systematic testing of combinatorial effects of different factors to induce differentiation of TECs at early stages and to establish an in vitro maturation protocol without grafting of TEPs in vivo. The current work systematically investigates the effects of many factors at three early stages (from DE to AFE to 3PPE to TEP) on the differentiation of human iPSCs into TEPs as well as the maturation of cortical and medullary TECs in hydrogel-based 3D co-cultures with primary hematopoietic progenitors ETPs. The authors could also show that ETPs differentiate into CD4 and CD8 T

cells and dendritic cells in the 3D hTO co-culture system. The experiments are well designed and the results add to the overall incremental understanding of the factors to differentiation induction at the different transition stages. Impressively, in the 3D hTO co-culture system, the iPS-derived TEPs continue to mature into functional TECs that support human ETPs to differentiate into T cells and dendritic cells.

Some comments are provided to improve the manuscript.

1. Fig. 1d and Supplementary Fig. 1d: It would be interesting to know how the authors used Plackett-Burman designs as a screening method to evaluate the most significant factors with the fewest experiments. For example, why is it necessary to have 36 runs for 5 factors on two levels (AFE: D5 – D7)? [70]

[70] The Plackett-Burman (PB) design is particularly suited for experiments where multiple factors need to be tested with a minimal number of runs, as is the case for iPSc to TEP differentiation due to workload and reagent costs. The primary constraint of this design is that the number of runs must be a multiple of four, with at least one more run than the total number of factors. While 36 runs are not strictly necessary for screening five factors at the initial AFE stage, a higher number of runs enhances statistical power and increases the likelihood of detecting small-effect factors reliably. However, in later stages of differentiation, increasing the number of runs becomes significantly more costly, as cultures must be maintained for an extended period. Thus, due to experimental constraints, we opted for 24 runs for the final 3PPE to TEP transition, despite the reduced screening power.

2. “However, we chose to keep BMP4 and FGF8 because of their consensual use to achieve 3PPE differentiation in previous studies^{37,39,40}.” “Similarly to above, we chose to keep FGF10 and EGF since we observed higher growth rate in cultures supplemented with EGF.” Given this, some of the key findings would warrant testing in a combination without these factors at the transition stages of 3PPE (D7 to D11) and TEP (D11 to D131) to determine whether they really add any significant benefit or can be eliminated to simplify the protocol. [71]

[71] DoE is a powerful methodology to simultaneously assess the effects of multiple experimental factors. Based on this, we advocate for a more systematic integration of statistics-based tools in iPSc differentiation protocol design. However, it is crucial to recognize the limitations of this approach, particularly when using low-power designs such as PB, which are incomplete plans optimized for cost-effective screening of key factors. Initially, the

differentiation protocol optimization was intended to rely entirely on DoE. However, empirical observations revealed effects of certain factors on cell growth that could not be captured by our experimental readout. Indeed, RNA-seq, which compares the transcriptome of differentiated cells to human pharyngeal developmental cell atlases, assesses the “quality” of differentiation but omits quantitative aspects. Notably, **new Supp Fig 4b** shows that EGF and FGF10 significantly promote cell growth, justifying their inclusion in our differentiation protocol. In addition, by assessing the transcriptome at each differentiation step, we assumed that optimizing cell purity at each stage would consistently enhance TEP generation later on. However, it is conceivable that a less pure intermediate differentiation product at the 3PPE stage could ultimately yield more TEPs at the end product of differentiation. Such effects may arise from intercellular signaling, where differentiation factors secreted by coexisting cell populations influence outcomes. Thus, we tested whether BMP4 and FGF8 combination, despite being neutral in our DoE, could enhance TEP generation when applied at the AFE-to-3PPE transition, as suggested by the literature. Using qPCR to directly quantify *FOXN1* expression at D15 as a measure of differentiation efficiency, we observed significantly higher *FOXN1* expression in TEP samples exposed to BMP4 and FGF8 at the AFE stage (**new Supp Fig 4a**). This shows that while these factors do not directly impact 3PPE differentiation efficiency, they enhance subsequent TEP generation, supporting their inclusion in the final protocol.

3. Lines 291-293: “We also identified the NOTCH ligands DLL1, DLL4, JAG1 and JAG2 whose coding genes show robust higher expression in TEP vs iPSc (Fig 3b, Right)”. Do the authors mean in CD205+ TEP vs TEP (or iPSC)? [72]

[72] We identified the NOTCH ligands JAG1, JAG2, and DLL4 as being robustly upregulated in the CD205⁺ TEP subpopulation compared to the iPSc from which they were derived. We also performed immunofluorescence to confirm the expression of DLL1 and DLL4 in TEPs (**new Fig 3e**). Thus, consistent with transcript differential expression, we demonstrate the membrane expression of these NOTCH ligands in the final differentiated TEPs.

4. Supplementary Fig. 6a: Please check the method section, should keep consistent, either using CD4/CD8 depletion or CD3 depletion, or both? [73]

[73] This was indeed a mistake, the method section has been corrected (**Line 820**): “and incubated with CD3 labeled Dynabeads (Thermofisher, 11031)”.

5. Supplementary method section is missing, please provide information on antibodies used in the study, also information and concentrations of factors used for TEP aggregation with ETPs and 2D co-culture, including RANKL, FGF10, IGF1, EGF, SCF, IL7, and FTL3L, lines 637-639. [74]

[74] All information and concentrations of factors used for TEP aggregation with ETPs and 2D co-culture have been added to the Method section, as well as the antibodies used to the reporting-summary file.

6. 2D co-culture and 3D hTO co-culture: please provide the ratio and cell numbers of TEPs and ETPs used. [75]

[75] For 3D hTO co-culture, 2,500 freshly sorted primary ETPs were seeded with 20,000 iPSc-derived TEPs during the reaggregation step in low binding 96-well plates. This information has been added to the Methods section. The same cell ratio was used for 2D co-cultures (**Line 371**).

7. Fig. 4: b, what are differences between left and right images? [76]

c, How is the co-localization quantified? [77]

d, What percentage of TEPs become mTECs? [78]

[76] There is no difference between left and right images. They are replicates from independent cultures under the same differentiation and co-culture supplementation conditions, using the same instrument and magnification. This highlights the consistent colocalization observed in 2D co-cultures, along with morphological changes in TEC-like cells after one week of co-culture with thymocytes. We have added this information to the figure legend of **Fig 4**.

[77] We used FIJI software to count T cells in images of 2D co-cultures, specifically those located within the membrane perimeter of adherent epithelial cells. We calculated the co-localization ratio by dividing this number by the total number of T cells. We have added this information to the figure legend of **Fig 4**.

[78] We used larger field IF images from the same differentiation to quantify differentiated cell populations based on *CLDN3* expression (**new Supp Fig 10b**). Park et al. have shown in their human thymic cell atlas that *CLDN3* expression is restricted to the most mature medullary compartments (mTECII - mTECIII) (**new Supp Fig 10a**). In contrast, *KRT14* and *KRT5* exhibit a broader mTEC expression pattern (**new Supp Fig 10a**). Quantification of *CLDN3*⁺ cells in

larger field IF images revealed that they constitute 14% of the total cell population, highlighting heterogeneity within the differentiated population and the low maturation of TEPs in 2D cultures. We have added this result to the main text (**Line 395**).

Fig. 4b, 5b, 6a, S1: please insert scale bars. [79]

[79] Scale bars have been added to the figures.

9. The discussion does not compare their hTO co-culture results to previously published work with grafting of TEPs in vivo for maturation induction of TECs and efficiency to support T cell and dendritic cell development. [80]

[80] Discussion has been amended (**Line 767**) with: "Ramos et al. reaggregated and grafted the TEPs they had differentiated into murine models to promote their maturation into TECs. They reported KRT5 and KRT8 positive cells but did not provide a quantitative assessment of the maturation level of the TEC products. In contrast, our study demonstrates the presence of a substantial HLA-DR positive epithelial population. However, the swine thymus graft proposed by Gras-Pena et al. successfully reproduced cortico-medullary segregation, possibly due to the pre-existing ECM structure, which may influence TEP fate decisions. In terms of T cell generation, both systems support thymopoiesis, with Ramos et al. generating both innate immune cells and mature T cells, while Gras-Pena et al. primarily observed DP cells. This highlights the closer resemblance of thymic grafts to the human thymus, whereas our ex vivo system exhibits lower DP and SP4 production. However, our system is more compatible with scalable, off-the-shelf therapies and carries a lower risk of rejection compared to swine thymus grafts."

We thank the reviewers for recognizing the major advances introduced in the R1 version and their constructive feedback which helped us further strengthen the new R2 version of our manuscript. We provide a detailed point-by-point response in blue for each reviewer below, and have highlighted the changes to the revised manuscript using tracked changes in red.

Comments to Reviewers:

Reviewer #3 (Remarks to the Author):

In the previous version of the manuscript “Combinatory differentiation of human induced pluripotent stem cells generates functional thymic epithelium driving dendritic- and CD4/CD8 T-cell full development”, by Provin, d’Arco, Giraud et al., we identified major points to address to justify statements made by the authors.

In this new revised version of the manuscript, the authors made significant improvements pertaining to the major suggestions for review, notably:

- Adding quantitative information (cell numbers) generated from iPSC (TEPs) and T-cell population differentiated in hTOs throughout the manuscript;
- Detailing the number of TEPs and progenitor cells needed for making hTOs;
- Clarifying the use of the DOE method;
- Verifying the reproducibility of the hTO by increasing the number of experiments;
- Adding data on the kinetics of T-cell development;
- Testing the efficiency of the hTOs in generating T cells when starting from HSPC not primed towards T-cell fates;
- A better characterization of TECs in hTO by showing gene and/or protein expression of markers of TEC identity and function (notably mTECs);
- A better characterization of non-T hematopoietic populations (DCs) and non-TEC stromal cells in hTOs;
- Confirming hTOs do not generate cells with NK and NKT fates;
- Improving figure legends, texts and revising statements based on new experiments.

Our minor points were also considered and addressed in this revised manuscript.

Two main issues remain in the current version:

1. There is no functional analysis of the T cells produced, likely because of low cell yield. The authors’ statement that this will be solved using future changes in methodology is rather overconfident and the issue should be directly addressed in the discussion as to why the output is so low. This is also an essential hurdle to overcome before translation is feasible and will also limit the experimental usefulness of the model.

We thank the reviewer for this important comment. We agree that the yield of mature T cells is currently limited and constitutes a hurdle for translational applications and broader experimental use. In our study, this limitation primarily reflects the restricted availability of hematopoietic progenitors from primary thymic samples, which constrained the number of organoids generated from each donor. Replacing primary ETPs with CD34+CD7+ iPSc-derived hematopoietic progenitors (HPSCs) is expected to substantially increase T cell yield, as TEP numbers from iPSc are not limiting, thereby alleviating the input constraint. We established aggregation of small TEP/ETP masses in 96-well plates, making our system inherently suited for scale-up. Seeding these aggregates within hydrogel wells will further benefit from automation to ensure robust and reproducible scalability. Hence, improving T-cell yield in our current hydrogel-based hTO system will rely on standardized iPSc-derived hematopoietic progenitors and automated organoid formation rather than changes in methodology per se. Within the same methodology, we could also tune gel stiffness and initial aggregate size to improve culture-medium diffusion, cell interactions within the hydrogel matrix, and ultimately enhance T-cell output. In addition to showing that hTOs support differentiation of CD34+CD7+ iPSc-derived HPSCs into T cells (Supp Fig 19-21), we have recently substantially improved the efficiency of the iPSc-to-HPSC protocol, ensuring that the input limitation in hTOs will be lifted and enabling the launch of large batches, further facilitated by automated seeding.

We amended the last sentence of the abstract (**Line 41** - " platform for studying thymic cellular interactions...") to indicate that optimization of cell production is the next step toward translational applications. We have added a new paragraph in the Discussion to address the reasons for the limited T-cell yield and to outline how, within our system, iPSc-derived HSPCs, automation and hydrogel refinement will improve T-cell production (**Line 766** - " At the current scale, T-cell output remains modest...").

2. A major point that should be corrected throughout the manuscript is of the nature of the seeding hematopoietic cells that can generate T cells in the hTOs. The only hematopoietic progenitors (whether from CB or iPSCs or thymus) able to generate T cells in the hTOs are those that express CD7(i.e. CD34+CD7+ cells). As mentioned by Reviewer 1, CD34+ CD7+ cells are not HSCs nor true thymic ETPs as described by the authors. In CB they are mostly lymphoid-committed with DC and NK potential. In the case of thymic progenitors, CD34+CD7+ cells are largely T committed lineage. True ETPs with multilineage potential are CD34+CD7- (Le et al, Immunity 2020 doi: 10.1016/j.immuni.2020.05.010). The inability of the hTOs to generate T cells from CD34+CD7- cells suggests that critical factors required for induction of T cell specification and commitment (most likely Notch signaling), present in normal thymus, are missing in the hTO microenvironment. This deficiency in the system is also seen in some (although not the OP9DLL4/4 or ATO) models and should be clearly acknowledged as such in the manuscript.

This is indeed an important point with two separate aspects:

The first point seems to be a matter of denomination. There is indeed a lack of nomenclature standardization across the literature for cells that are about to or have just entered the thymus. Some studies, including Le et. al. (Immunity 2020), define CD34+CD7- cells that are about to enter the thymus as "ETP", while other human studies functionally and transcriptionally characterize these cells as "TSP1", and refer to them as "TSP2" upon CD7 acquisition, using

the “ETP” denomination for the heterogeneous population of intrathymic multipotent CD34+CD7+CD44+ cells encompassing the DN1 stage, before they become fully committed to the T lineage at the DN2a/b stage, marked by CD44 downregulation (Carpenter et al. *Nature Immunology* 2010 - PMID:20644572; Canté-Barrett et al. *Front Immunol* 2017 - PMID:28163708; Lavaert et al. *Immunity* 2020 - PMID:32304633; Ling Liang et al. *Nature Immunology* 2023 - PMID:36703005)). In our study, we adopted the latter nomenclature, using “ETP” to designate intrathymic CD34+CD7+CD44+ cells, and have added the three most recent references cited above to support the sentence clarifying this point in the main text (**Lines 122** - “...progeny at the double negative (DN)1 stage”).

The second matter concerns the differentiation potential and commitment status of CD34+CD7+ cells. The aforementioned studies, including Le et. al. (*Immunol* 2020), show that primary CD34+CD7+CD44+ cells are not committed to the T lineage and retain multilineage potential, with the capacity to give rise to T, DC/myeloid, NK, B and erythroid cells *in vitro*. Upon exposure to Notch signaling in the thymus, these progenitors preferentially differentiate into T cells and DCs, precisely the cell types we observe in our hTO system, highlighting the role of Notch in driving these cells towards T cell commitment.

iPSc-derived CD34+CD7+CD44+ cells display the same multilineage potential and Notch-restricted T and DC cell fates in our system, undergoing T-lineage commitment and demonstrating that our hTO system supports T cell commitment from multilineage CD34+CD7+CD44+ progenitors. It however does not support the induction of T-lineage differentiation from CD34+CD7- cells, likely due to the absence of a strong enough Notch signaling burst required for initial induction (Strubbe & Taghon (*Biochem. Soc. Trans.* 2021 - PMID:34643218)). However, Notch levels in TEPs appear sufficient to support progression from uncommitted CD34+CD7+CD44+ progenitors to the DN2 stage, where T lineage commitment becomes irreversible.

Reviewer #5 (Remarks to the Author):

The authors have addressed all of my previous comments; however, there are still some issues that require further clarification. In particular, the authors have added new experiments to support their conclusions.

1. Please avoid using the same abbreviation, DE, for both "definitive endoderm" and "differentially expression" in the manuscript, as this may cause confusion.

Thank you for highlighting this point. We kept DE as the abbreviation for "definitive endoderm" and chose not to abbreviate "differential expression", as it occurs less frequently in the manuscript and figures, to avoid confusion.

Additionally, please carefully review the manuscript including the figures for grammatical errors and typos, for example, "differentially expression", "characterization", "PAX9 et FOXN1", "Protocol reproductibility", "An 3D thymic", "FTL3L", "reagregation", "HLA-DRhi et lo", "pharyngealn"

Thank you for pointing out examples of grammatical errors and typos. We have corrected the cited ones, and others, in carefully reviewing the entire text, legends and figures.

2. Please provide the gene lists for the different reference cell populations used in the comparison of iPSC-derived samples to the two atlases. Additionally, include the GEO accession codes for the DGE-seq data from iPSc to TEP.

We provide two Supplementary Tables (**Supp Table1** and **Supp Table2**) listing the reference cell populations from the Han and Magaletta scRNA-seq atlases, for which we computed specific marker genes and reported the corresponding statistics, including effect size.

We also completed uploading the DGE-seq reads to NCBI for the DOE-based factor screening (GSE302942 - *token:kfkfuiwoltlbyf*) and for the time-course of iPSc differentiation to TEPs (GSE304598 - *token:olavokwqjzkvzsd*).

This adds to the previously uploaded datasets:

GSE294118: CD205+ iPSc-derived thymic epithelial progenitor cells - *token:sfstsgcoptgfvz*

GSE293928: Human iPSc-derived thymic organoids (hTOs) seeded with primary early thymic progenitors (ETPs) at early phase of maturation (D7)- *token:epuruwembrsfduz*

GSE294061: Isolated TEC and DC from human iPSc-derived thymic organoids (hTOs) seeded with primary early thymic progenitors (ETPs) at late phase of maturation (D28) - *token:ofypiqwyvvotpmh*

GSE293930: Detaching cells from human iPSc-derived thymic organoids (hTOs) seeded with primary early thymic progenitors (ETPs) at late phase of maturation (D28) - *token:mvuygikpfsxdiz*

3. **Supplementary Fig. 4b:** Are the proliferating cells in the differentiation end product TEPs or another cell type? How is the TEP population defined here? What percentage of the differentiation end product are TEPs?

In Supplementary Fig. 4b, as in other panels, the TEP population is defined as the end product of our DOE-optimized iPSc-to-TEP differentiation protocol. Our intent was to monitor the proliferative behavior of the differentiation output under EGF and FGF10 supplementation. We revised the Supplementary Fig. 4 legend to clarify that TEP refers to the differentiation end product.

We understand the reviewer’s concern to ensure that the observed increase in proliferation under EGF and FGF10 supplementation, applies to bona fide TEPs (EPCAM+ CD205+). In this experiment we quantified cell numbers by FACS and included EPCAM and CD205 staining. We thus could assess the efficiency of differentiation into CD205+ TEPs, which was stable across conditions (68%- 79%) and within our expected experimental range as shown in Fig. 3a (Right).

Percentage of EPCAM+CD205+ cells in differentiation end product cells. Flow cytometry quantification at the end of iPSc-to-TEP differentiation. Barplot shows the proportion of EPCAM+CD205+ cells among total cells under the indicated EGF/FGF10 supplementation conditions.

EGF	-	+	-	+
FGF10	-	-	+	+
Number of end product cells per MW6 well (mean of 2 replicates)	5.975.10 ⁶	5.775.10 ⁶	6.4.10 ⁶	7.10 ⁶
Number of CD205+ TEPs per MW6 well (mean of 2 replicates)	4.356.10 ⁶	4.539.10 ⁶	4.716.10 ⁶	4.739.10 ⁶

To better illustrate the proliferative dynamics of the end product cells and the CD205+ TEPs, we normalized cell counts to the baseline (EGF- FGF-) condition. This shows that both cell populations expand under EGF and FGF stimulation, although the combined effect of the two factors appears less marked for the CD205+ TEPs.

Relative cell number of differentiation end product cells and EPCAM+CD205+ TEPs. Flow cytometry counts at the end of iPSc-to-TEP differentiation, normalized to the EGF-/FGF10-condition (set to 1). Lines show total end product cells (blue) and EPCAM+CD205+ TEPs (orange) across the indicated supplementation conditions. (n = 2 replicates).

Considering Fig. 2a and Supplementary Fig. 6 (modules 4 and 6), a set of genes related to muscle development appears to be highly expressed in the differentiation end product (D14-D22). Could you provide an explanation for this observation?

Regarding the expression of genes related to muscle development, we suspect it results from the presence of a cardiac differentiation by-product among mis-differentiated cells, as achieving a pure differentiation product is challenging. Cardiac phenotypes are a common contaminant as their differentiation from iPSc involves regulation of the Activin/Nodal, Wnt and BMP4 pathways (among others) (Batalov & Feinberg, Biomark Insights 2015 - PMID:26052225), which are involved in many differentiation processes, including our iPSc-to-TEP differentiation protocol.

4. Supplementary Fig. 5b: It would be informative to include a comparison between primary TECs and D14-D22 iPSc-derived samples.

The comparison between primary TEC and TEP samples was performed using the same DESeq2-based script as for the primary TECs vs iPSc comparison. Results were visualized with an MA plot added in the **revised Supp Fig 5**. Genes upregulated in TECs include classical maturation markers such as HLA-DRA, HLA-DRB1, HLA-DQB2, and members of the CCL (CCL3, CCL19, CCL25) and CCR (CCR9) families, whereas TEP-enriched genes comprise members of the PRSS (PRSS16, PRSS23) and Integrin (ITGA5, ITGA6, ITGAV) families, as well as KRT19. This is consistent with the acquisition in TECs of a thymic functional program, distinct from the immature epithelial state of TEPs. This has been added to the main text (**Line 254** - " Furthermore, a direct comparison between primary TECs and TEPs..."). Note that a skew toward TECs is observed among low-count genes, which may reflect technical noise or residual normalization bias after shrinkage.

5. What percentage of cells are double positive for FOXN1 and PAX9 (**Supplementary Fig. 7a**) in the differentiation end product?

PAX9 is a transcription factor associated with anterior pharyngeal endoderm patterning, with peak expression in the 3PPE, where it contributes to the specification of thymic epithelial progenitors. Its expression remains detectable in TEPs, marking pre-thymic epithelial identity. FOXN1, in contrast, is progressively upregulated during thymic epithelial lineage commitment and is essential for subsequent TEC differentiation and maturation. Accordingly, some degree of co-expression is expected at the onset of lineage engagement, although PAX9 is not required for FOXN1 induction nor for TEC commitment per se. Our objective was to confirm PAX9 expression to underscore the pre-thymic identity of our TEPs, and, on the other hand, FOXN1 expression to confirm their commitment toward the mature thymic epithelial lineage. We optimized the detection of each marker separately, selecting the most robust antibodies, which both turned out to be rabbit-derived, thereby precluding co-staining. We have revised the main text to specify that PAX9 marks early pharyngeal (pre-thymic) identity, while FOXN1 indicates thymic epithelial lineage commitment and maturation (**Line 274** - "In addition, we showed that these cells express PAX9..."), thereby clarifying that although both are expressed to some extent in TEPs, their expression dynamics remain distinct.

6. **Fig. 3c and line 334**: The manuscript states that "one third of these cells with inferred TEC signature are medullary oriented", how was this proportion calculated? In **Fig. 3c**, the fraction appears to be less than 20%.

There was indeed confusion when interpreting the results from our deconvolution analysis, as we mistakenly wrote the value from the cTEC population in the sentence instead of the one from the mTECs. The figure stated here represents the mean of the deconvolution estimation (percentage of cells with target population identity) in TEP samples. The exact numbers are 28.75% for cTECs and 17% for mTECs. The main text has been corrected to be coherent with the results (**Line 314** - " We then performed a deconvolution analysis against the Magaletta dataset...").

7. **Fig. 3d and line 335**: The manuscript states, "Moreover, classic cTEC markers are less expressed in the CD205+ TEPs". Less expressed compared to what population? This statement is unclear and needs further clarification.

The sentence was unclear and has been restructured to emphasize that CD205+ sorted TEPs show lower expression of cTEC markers than unsorted TEPs. "Moreover, comparison of classical cTEC marker expression between CD205+ sorted and unsorted TEPs showed lower expression in the CD205+ TEPs (**Fig. 3d**)" (**Line 316** - "Moreover, comparison of classical...").

8. **Lines 336-337**: the manuscript states, "identified NOTCH ligands DLL1, DLL4, JAG1 and JAG2, whose coding genes exhibit robustly higher expression in TEPs compared to iPSc (Fig. 3b, Right)." However, in Fig. 3b (Right), these genes appear to be expressed at lower levels in TEPs, with the exception of JAG2. As noted in my initial review, the authors may be referring to CD205+ TEPs, but in this case, DLL1 expression appears comparable to, or even lower than that in iPSCs, not higher. This should be clarified.

Thank you for pointing out this discrepancy. The text has been modified to specify “CD205+ TEPs” and to reflect the high DLL1 expression levels in iPSc (**Line 318** - " We identified several NOTCH ligands..."). DLL1 expression is expected in iPSc, as it contributes in the balance between self-renewal and differentiation in pluripotent stem cells, promoting stemness maintenance through pluripotency marker expression and blocking differentiation toward multiple lineages (Fang et al, Journal of molecular and cellular cardiology 2019 - PMID:31233755; Ivey et al, Cell Stem Cell 2008 - PMID:18371447).

Reviewer #6 (Remarks to the Author):

While the authors have been largely responsive to reviewer 1's concerns, there are still a few questions that remain, especially about the maturation and development of thymocytes within the organoid system. This group has done a commendable job on generating what look to be organoids that resemble thymic epithelium in many ways. The major claim of this manuscript is that unlike other approaches that require an in vivo transplantation step, this is a fully ex vivo organoid able to generate mature T cells. However, I am not yet fully convinced that is what they are doing.

Firstly, CD4/CD8 expression seems to follow a curious pattern. On **Line 566-568** the authors claim that there are CD4⁺CD8⁺ double positive cells (Fig. 7D) but that is not apparent from the flow plots being presented (a point that was raised in the prior reviewer comments).

We realized that we had not formally responded to this point in the previous revision. We have now modified this sentence by: "Although most CD3⁺ cells are CD4⁻CD8⁻ DN thymocytes, we found that hTOs also contained CD4⁺CD8⁻ SP4 and CD4⁻CD8⁺ SP8 cells, albeit in lesser proportions (Fig. 7d)." (**Line 536** - "Although most CD3⁺ cells are CD4⁻CD8⁻ DN...").

and added: "Although CD4⁺CD8⁺ double-positive (DP) cells were not observed, other hTO experiments, albeit at similar timepoints, have shown their presence at low frequency within the CD3⁺ fraction (Fig. 8a,b)" (**Line 538** - "Although CD4⁺CD8⁺ double-positive (DP)...").

In **Fig. 8** there are more DP cells, but as far as I can gather the reason why is that in Fig. 8 there is a broader CD45⁺ gate, not excluding CD3⁻ cells. Based on this it seems that most DP cells are CD3⁻. This is surprising given the proportion of CD3⁺ cells that seem to be generated. In **Fig. 8a** it seems that there are a lot of both CD4⁺ and CD8⁺ single positive cells that are also CD3⁻ (and very few that are CD3⁺). Finally, it also seems that most CD3⁺ cells are not TCRab or gd⁺. Based on the argument that the abundant CD3⁺TCR⁻ cells are DN cells expressing preTCR (which would explain the expression of CD3 even early in the cultures), this would suggest downregulation of CD3 between the preTa stage and DP (and later) stages? Ideally, characterization of these earlier DN stages of T cell development using markers such as CD34, CD7 and CD5 would reveal what is happening within the DN compartment with respect to this unexpected expression of CD3.

We thank the reviewer for raising up these points whose clarification will help improve the manuscript.

Firstly, we modified Fig 8a to clarify that the CD45⁺ CD3⁺ fraction displayed for each of the five hTO experiments was derived from the CD45⁺ gate, and added stacked bars to show the percentage of CD3⁺ cells within each DN compartment (**revised Fig 8a**). We observe a rising proportion of CD3⁺ cells ranging from 4% to 50% in the late D38 hTO.

To get insight into the DN compartment, we followed the reviewer's recommendation and took advantage of the scRNA-seq we performed on CD45⁺ cells from the D28 hTO (mid-course) of Fig 9.

Please note that in the revised version of Fig. 9f, we applied a projection of our data onto Park's reference rather than integration, since all our cells map to the reference clusters. Projection preserves the structure of the reference clusters, especially where query cells are abundant, by avoiding mutual correction between the reference (Park) and the query (our CD45+ cells). This ensures a more accurate and topologically consistent mapping of our CD45+ cells. Although we mentioned in the corresponding part of the results "projection", the previous Fig. 9f actually showed integration. We corrected this mistake in the revised Fig. 9f and Methods (Line 998 - "For the integration, anchors were computed with...").

Projection of hTO scRNA-seq onto a human thymic reference. (Left): UMAP of the reference human thymic atlas (Park et al.) with annotated clusters (cTEC, mTEC, DC1/2, DN, DP, SP, etc.). (Right): projection of dissociated hTO scRNA-seq cells (black) onto the same reference embedding (reference cells in grey); labels indicate the reference identities used for annotation.

We selected the CD45+ cells that mapped to the DN, SP, DC1 and DC2 clusters, and assessed the expression of *CD34* and *FLT3*, which both are markers of uncommitted DN cells. Neither was expressed in DN, except for *FLT3* in DCs where its expression is expected, supporting that DNs have undergone T cell commitment. This has been added to the main text (Line 616 - "It also identified DN thymocytes, which showed absence of *CD34* and *FLT3*...") and to the new Fig. 9g).

***CD34* and *FLT3* expression in D28 hTO scRNA-seq.** Violin plots of normalized expression for *CD34* (left) and *FLT3* (right) in CD45+ cells projected to DN, SP, DC1, and DC2 clusters. Each dot represents one cell; *n* indicates the number of cells per cluster.

We also showed that the DNs expressed *CD3* transcripts, based on a gene module comprising *CD3D*, *CD3E*, *CD3G* and *CD24*. In contrast, DCs showed no expression of this module and served as a negative control. In addition, DNs expressed genes encoding key pre-TCR/TCR signaling components (*LAT*, *ZAP70*, *LCK*) and the signal-responsive *CD5*, grouped into a second gene module that remains silent in DCs. These new findings confirm that DN cells have passed T cell commitment and are consistent with assembly of the rearranged TCR beta chain with the pre-alpha chain and *CD3*, the latter likely remaining cytoplasmic (*CD3*- DN) or present at the cell surface (*CD3*+ DN), as shown in other hTO experiments by FACS analysis. This has been added to the main text (**Line 616** - "It also identified DN thymocytes, which showed absence of *CD34* and *FLT3*...").

Expression of TCR signaling/response and CD3 gene modules in D28 hTO scRNA-seq. Violin plots of normalized expression of a TCR signaling/response module (*LAT*, *ZAP70*, *LCK*, *CD5*; left) and a CD3 gene module (*CD3D*, *CD3E*, *CD3G*, *CD247*; right) in *CD45*+ cells projected to DN, SP, DC1, and DC2 clusters. SP violins are displayed with transparency to highlight expression patterns in DN and DC subsets. Each dot represents one cell.

In Fig. 8a, (D27 hTO), we observe a large proportion of *CD4*+ cells and *CD4*+*CD8*+ cells that are mostly *CD3*- (present in the *CD45*+ compartment but largely absent from the *CD3*+ fraction). This suggests that these cells have progressed beyond the ISP stage (without yet upregulating surface *CD3*) to a *CD3*- DP stage, but rarely complete the transition to *CD3*+ DP and subsequent *CD3*+ SP4 and SP8. These results point to a bottleneck at the *CD3*- DP step, likely due to limiting interaction of the *CD3*- DP cells with Notch-ligand-positive cTEC at this stage of hTO formation.

In D31 hTO, we observe a *CD3*- versus *CD3*+ pattern similar to, but less marked than, that in D27 hTO, with a smaller proportion of *CD3*- cells in the DN fraction, indicating accumulation of *CD3*+ DN cells. This accumulation is even stronger in later hTOs, reaching ~50% of the DN compartment, and is accompanied by a decrease in *CD3*- *CD4*+ (ISP) (Staal et. al. Stem Cells 2001) and *CD3*- DP, consistent with a block at engagement to the ISP stage. Together with the constant production of *CD3*+ SP4 and *CD3*+ SP8 cells throughout hTO formation (Fig 8c, Right), these data suggest that the more committed DN cells mature directly into *CD3*+ DP at a limiting yield and are then rapidly selected into *CD3*+ SP8 and SP4 cells, leaving little intermediate accumulation of DP within the *CD3*+ fractions. We modified the main text (**Line 564** - "before surface *CD3* upregulation, but only...") and amended the Discussion (**Line 734** - "However, the underrepresentation of DP cells and the...").

Hence the large proportion of TCRab⁻ and TCRgd⁻ cells within the CD45⁺ CD3⁺ compartment (Fig. 7d) consists mainly of DN cells whose net abundance likely results from DN proliferation and their limited progression to the DP stage. We amended the Discussion (**Line 744** - “We also observed that a substantial...”)

There also seems to be significant variation in outcomes with **Fig. 7a** showing 17% CD3⁺ at day 21 but **Fig. 7c** showing 47% (this is reflected by the graph in Fig. 7b which shows a very large spread of CD3⁺ cells at day 21).

As pointed out by the reviewer, there is significant variation in the proportion of CD3⁺ cells at D21 and in later hTOs (Fig. 7b). Because most CD3⁺ cells are DN (Fig. 8c), the CD3 proportion variation is observed in the DN compartment (**revised Fig. 8a**, *Stacked bars*). It likely arises from differences in hTO development with CD3⁺ DN cells accumulating at later stages in some organoids. This point has been added to the main text (**Line 527** - “We also note substantial variation...”)

Is there a difference in TCR expression in the very high CD3 expressing organoids vs the low CD3 expressing?

We addressed this question at the level of SP T-cell generation rather than TCR expression, since TCR antibodies were not systematically included in the FACS panels used to track SP T-cell output. Generally, we observed a correlation between total CD3⁺ generation or CD3⁺ DN generation (the former reflecting the latter) and the production of CD3⁺ SP8 or CD3⁺ SP4 cells (Fig 8c, Right). This indicates that the more CD3⁺ DN cells are observed, the more CD3⁺ SP cells emerge, consistent with a constant but limited progression to SP stages.

To test whether the correlation between CD3⁺ and SP T-cell generation is also observed at comparable developmental days in hTOs with differing CD3⁺ levels, we analyzed four hTO experiments (two D34 and two D35). CD3⁺ SP8 output scaled with CD3⁺ DN abundance, whereas the correlation was weaker for CD3⁺ SP4, possibly reflecting their reliance on MHCII-expressing mTECs (mTEC^{hi}) with access to CD4-supporting niches and the timing of their maturation varying between organoids.

CD3⁺ SP8 vs CD3⁺ DN cells per organoid in D34 and D35 hTOs

CD3⁺ SP4 vs CD3⁺ DN cells per organoid in D34 and D35 hTOs

CD3⁺ SP8/SP4 vs CD3⁺ DN per organoid (D34-D35 hTOs). Scatter plots of absolute cell numbers per organoid quantified by flow cytometry. Left: CD3⁺ SP8 cells vs CD3⁺ DN cells. Right: CD3⁺ SP4 cells vs CD3⁺ DN cells. Four hTO experiments (two D34, two D35). DN = CD4⁻CD8⁻; SP8 = CD4⁻CD8⁺; SP4 = CD4⁺CD8⁻. Each dot represents one organoid.

In **Fig. 7D** the authors would like to claim differentiation to a mature CCR7+CD62L+ phenotype, but are these cells TCR+ or just CD3+? Ultimately, what evidence do the authors have that there are a significant fraction of CD3+ cells have actually undergone TCR recombination with selection? RAG induction? CD5 signaling? TRECs? Are they generated but is it just an extremely inefficient process? The biggest innovation of this study is the ex vivo generation of mature T cells in thymic organoids – even if in small numbers to start - so ensuring the robustness of this claim is important. Furthermore, the authors do not seem to have fully addressed these same questions about TCR expression from Reviewer 3 in their response.

The CCR7+ CD62L+ SP cells observed in hTOs (Fig. 7d) are indicative of thymocytes that have acquired a mature TCR, since this CCR7+ CD62L+ phenotype is restricted to SP cells with a functional TCR engaged in thymic selection and ready to emigrate from the thymus. Although direct antibody-based detection was not possible since TCR antibodies were not included in the FACS panel, we used transcriptomic data from the D28 hTO scRNA-seq. We compared the TCR-signaling/response gene module (LAT, ZAP70, LCK, CD5) between hTO cells mapping onto the SP and DN clusters of Park et al., and found it significantly upregulated in SP cells. Similarly, CD3 gene module expression was significantly higher in SP cells, as expected once the alpha chain has rearranged and assembled with the beta chain to form a mature TCR. These findings support that the SP cells generated in hTOs carry functional TCRs, in line with results from iPSc-derived HSPC hTOs in which all generated SP8 are TCR+. Together, these data show that the SP cells upregulate both TCR signaling/ response and CD3 gene modules, consistent with acquisition of mature TCRs. This has been specified in the main text (**Line 621** - “In SP cells, we observed a further significant upregulation...”) and added to the **new Fig. 9g**.

Expression of TCR signaling/response and CD3 gene modules in D28 hTO scRNA-seq. Violin plots of normalized expression of a TCR signaling/response module (*LAT, ZAP70, LCK, CD5*; left) and a CD3 gene module (*CD3D, CD3E, CD3G, CD247*; right) in CD45+ cells projected to DN, SP, DC1, and DC2 clusters. Each dot represents one cell. Statistical comparison of DN vs SP: $p = 0.025$ (TCR module), $p = 0.040$ (CD3 module) (Welch’s t-test, one-sided).

To further address TCR recombination, we examined whether developing thymocytes in the same scRNA-seq dataset were in a permissive state for TCR chain gene rearrangement. We analyzed the V(D)J recombination gene module (*RAG1, RAG2, DNNT*) which encodes the

core enzymes required for V(D)J rearrangement. Among 116 cells in our dataset, only three expressed this module unambiguously: the two DP cells present and one additional cell. We highlight this point here because the probability that the two DP cells would be positive by chance is less than 1 in 2,000 ($p < 0.00045$ hypergeometric test). However, since only two DP cells were captured, we did not include this result in the revised manuscript, although the statistics make the observation reliable.

We used a hypergeometric probability:

$dhyper(\text{number of RAG module+ cells in the DP cluster, number of RAG module+ cells, total number of cells} - \text{number of RAG module+ cells, size of the DP cluster})$

$dhyper(2,3,116-3,2): P=0.0004497751$

TCR_VDJ_Recombination
RAG1, RAG2, DNNT

Expression of the V(D)J recombination gene module in D28 hTO scRNA-seq. Violin plot of normalized expression for the TCR V(D)J recombination module (*RAG1*, *RAG2*, *DNNT*) in CD45⁺ cells projected to DN, DP ($n = 2$), SP, DC1, and DC2 clusters. Each dot represents one cell.

The absence of expression of the V(D)J recombination gene module in DN cells is consistent with their robust CD3 module gene expression, reflecting prior beta chain rearrangement and subsequent RAG gene downregulation to prevent premature alpha chain recombination (Kuo & Schlissel *Curr Opin Immunol* 2009 - PMID:19359154; Monroe et al. *Immunity* 1999 - PMID:10485655). In contrast, reactivation of the V(D)J recombination gene module in DP cells aligns with the expected TCR alpha rearrangement and assembly of a mature TCR, before being downregulated again at the SP stage.

In **Fig. 8c** the authors present kinetic data but there is no change in proportion of SP4 or SP8 cells from a very early timepoint of day 10. Do the authors think there are mature T cells that early and then do not change in proportion (despite the fact that no new hematopoietic progenitors are being seeded) across the timecourse of the organoid? This does not seem to fit with what we know about T cell development kinetics in the thymus and what could be expected to happen to SP proportions once earlier precursors are depleted.

We observe at day 10 CD3⁺ SP4 and SP8 cells that have passed positive selection, although they are unlikely to have completed later stages of selection and reached full maturation. This time window is shorter than the complete DN-to-SP transition that typically spans 2–3 weeks in adult murine thymus under steady-state conditions. However, several experimental systems demonstrate that this transition can occur more rapidly under synchronized or permissive conditions. Indeed, in the SCID-hu model, intrathymic injection of human fetal CD3⁺ 4- 8- thymocytes, of which more than 50% are CD34⁺ DN cells, leads to the emergence of CD3⁺ SP4 and SP8 cells within 4 days (Kraft DL *J.Ex.Med* 1993 - PMID:8315382). In a murine

context, intrathymic transfer of fully uncommitted DN1 thymocytes (CD44⁺CD25⁻, 14 dpc) produces CD3⁺ SP4 and SP8 cells by day 12 (Douagi I Eur. J. Immunol. 2000 - PMID:10940911). Ex vivo, FTOC with the same DN1 population yields SP4 and SP8 cells between days 9 and 11. These data show that pre-commitment thymocytes can generate CD3⁺ SP4 and SP8 cells within 10–12 days in optimized systems, supporting our observation in hTOs. This has been added to the main text (**Line 573** - “Note that CD3⁺ SP4 and SP8 cells were already detectable...”)

In the hTOs, we observe over time (Fig. 8c, Right) an increase in the absolute numbers of CD3⁺ SP8 and SP4 cells, more pronounced for CD8⁺, and an even more marked increase of CD3⁺ DN cells. This sustained DN accumulation reflects DN proliferation and survival despite the fact that no new hematopoietic progenitors are being seeded over time. Consequently, CD3⁺ DN cells become predominant both in frequency and absolute number, accounting for their high proportions at the different hTO time points (Fig. 8c, Left), and masking the increase in CD3⁺ SP cells, particularly CD3⁺ SP4 and DP cells, whose relative proportions decline over time despite absolute numbers increasing. Please also note that we chose a log scale representation to better accommodate the dynamic range of the different subsets within a single graph, which may visually amplify changes in lower-proportion populations. The main text has been modified (**Line 584** - “CD3⁺ DN thymocytes, thus masking...”)

Finally, I would suggest that the authors could again spend some time working on the wording in the manuscript (including greater experimental detail in the figure legends) to make it easier to follow.

We thoroughly revised the main text and revised the figure legends to increase accuracy and provide additional experimental context, making the figures clearer.

In addition, while I applaud the inclusion of a lot of validation data, 22 supplemental figures seems excessive and adds to the complexity. Including only the truly pertinent supplemental data may assist in the flow of the manuscript.

We acknowledge that 22 supplementary figures may appear excessive. We originally included 8 and added 14 to present the full set of additional results requested by reviewers 1, 3 and 5 (revision R1). Among these figures, we believe that **Supplementary Fig. 15** (Gating strategy for sorting CD3⁺ and CD45⁺ cells from hTOs), **Supplementary Fig. 16** (Gating strategy for sorting CD3⁺ cells from hTOs) and **Supplementary Fig. 17** (Unstained controls for CD4 and CD8 from hTOs), could be removed as they show gating strategies and negative controls, and do not provide essential information to the manuscript.

We thank the reviewers for their input, comments and positive feedback. We provide a detailed point-by-point response in blue for each reviewer below.

Comments to Reviewers:

Reviewer #3 (Remarks to the Author):

Most important issues are now addressed. However I continue to believe that the conclusions of how well the hTO can recapitulate normal thymopoiesis should be more circumspect. Given that the hTO model was only shown to generate T cells from primary thymic or iPSC-derived CD34+CD7+ cells, and not CB CD34+CD7- cells it is clear that T cells cannot be produced from uncommitted HSC or HSPC in the hTO system (unlike other in vitro systems). This very likely because of insufficient notch ligand expression in the iPSC derived epithelial cells, possibly because of immaturity. The poor output of CD4+CD8+ (DPs) also suggests that T cell differentiation is not proceeding normally (as suggested by reviewer #6). The statement (line 776) that “T-cell output remains modest mainly because the number of hematopoietic progenitors available from primary thymic samples restricts input and thus limits the size of organoid batches” begs the question of why the hTO system only worked with thymocyte progenitors.

While an alternative approach to produce TECs from iPSCs is a valuable addition to the literature I believe that it is important for the readership to understand the limits to the conclusions that can be made from the data (modifications should be made for example: line 663 “...functional thymic organoids supporting full thymopoiesis in vitro”)

I would also suggest removing the new sentence at the end of the abstract (line 41-42) “...generating mature thymic immune cells of clinical relevance and, through optimization of their production, paving the way for future cellular therapies.”

-It would seem that the clinical relevance of the cells and their optimization of production are yet to be explored and should be made clear are speculative at this point.

We thank you for your thorough examination of our work and for your valuable input throughout the reviewing process.

We agree that the requirement for CD34+CD7+ progenitors, as reported in the results, likely reflects Notch ligand levels in our system that can sustain T-lineage progression, but not the stronger burst required to drive initial specification from CD34+CD7- cells. This will guide future optimization.

In accordance with your latest remarks, we have introduced the following changes to tone down our conclusions:

Title: “full” was removed

Line 517, “Thymopoiesis” was replaced with “T cell differentiation”: **Importantly, we observed a CD3⁺ population arising at D17 and rapidly expanding at D21, reaching up to 60% of the hematopoietic population at D35, confirming active T cell differentiation in our system (Fig. 7a-b).**

Line 654, “full” was removed from the sentence

Line 731, “confirming the ability of hTOs to support full thymopoiesis” was removed: Interestingly, flow cytometry confirmed the ability of our system to generate both $\alpha\beta$ and $\gamma\delta$ T lineages, in contrast to studies showing an abnormal restriction to $\gamma\delta$ T generation.

Line 740, “Thymopoiesis” was replaced with “T cell differentiation”: Given recent reports highlighting the critical role of intrathymic DCs in complementary thymocyte selection, this population may contribute to T cell differentiation in hTOs, potentially promoting commitment to the SP CD4⁺ fate.

Regarding the clinical relevance of the cells produced within our hTOs and perspectives towards clinical applications, the following modifications were made:

Abstract, line 31: This sentence has been reworked and now points to new research perspectives towards therapeutic applications, supporting the idea that more research and optimization are needed before clinical applications can be considered.

New sentence: Thus, the presented thymic organoid model provides a practical platform for studying thymic cellular interactions and thymopoiesis in vitro, and opens new research perspectives towards cell-based therapies.

Line 766: The passage “With the aforementioned necessary improvements and additional studies” was added at the beginning of the sentence: With the aforementioned necessary improvements and additional studies, we foresee two main clinical applications for this system.

Line 780: This sentence was reworked in the same manner as the abstract, shifting the focus towards new research perspectives rather than cell therapy applications.

Former version: The use of iPSc presents promising long-term perspectives for regenerative medicine and cellular therapies, particularly for the in vitro generation of engineered T lymphocytes.

New version: The use of iPSc opens promising research perspectives for long-term applications in regenerative medicine and cellular therapies, particularly via the in vitro generation of engineered T lymphocytes.

Reviewer #4 (Remarks to the Author):

We thank you for your contribution to this reviewing process!

Reviewer #5 (Remarks to the Author):

In the new revised manuscript, the authors have thoroughly addressed all of my comments, I have no further questions or concerns.

We thank you for your comments and input throughout this reviewing process!

Reviewer #6 (Remarks to the Author):

Let me begin by saying that I like the approach the authors have taken and they have done a commendable job to advance the differentiation of functional TECs from pluripotent cells. I also appreciate the authors have tried to do as much as they can with their existing data to answer my questions and concerns – and I do genuinely try to avoid asking for experiments where they are not always needed (especially having been brought in after another reviewer dropped out). However, I am still concerned that the data presented are not entirely convincing that there are truly TCR rearranged and functionally expressing T cells being generated (let alone being MHC restricted and functional). Ideally, I would have loved to see additional data such as TRECs, dual staining for both CD3 and TCR, and even a functional measure of TCR engagement; however, I appreciate that may be beyond the scope of the manuscript and do not feel that these concerns should hold up publication.

Further digging suggests that, unlike mouse equivalents that typically detect only single TCR chains, the TCR antibodies used do seem to bind both alpha and beta chains (or gamma and delta chains) so should be detecting fully rearranged TCR (and not pre-TCR rearranged beta chain for instance). This should be highlighted as evidence of full TCR rearrangement. Progress has certainly been made in these systems, and it would be beneficial for the community to have these studies published, but ultimately they suggest that there is extremely inefficient generation of mature T cells.

I appreciate that the discussion does acknowledge a likely bottle neck between DN and DP, but to address my concerns, I suggest that the authors should consider working on some of the language to more carefully describe what is being generated and the limitations as they stand. For instance, the title referencing “full T cell development” seems a little premature since there is no functional evidence that these are indeed fully mature T cells. In the abstract, perhaps the language could be softened (further) to suggest that further optimization will be needed as the generation of mature T cells is extremely inefficient and the cells generated have not yet been tested functionally.

We thank you for your valuable questions and inputs, and for stepping in to help us complete the reviewing process!

Many of your suggestions regarding the functionality of the T cells and of the TCR fell indeed a little past the scope of this manuscript but are very present in our minds and are part of our goals for the continuation of this work. Although we don't show CD3 and TCR staining on the same plot, please note that our TCR staining plots are either gated within CD8+ cells that are themselves gated as CD45+ CD3+ cells (supplementary figure S21), or gated on CD3+ cells directly (figure 7c).

Following your suggestion, we have added a clarification in the Results section (line 523) noting that the anti-human TCR $\alpha\beta$ and TCR $\gamma\delta$ antibodies used in our study, unlike their murine equivalents, detect fully assembled heterodimeric receptors, which confirms complete V(D)J recombination of both chains and supports that these cells have indeed progressed beyond the pre-TCR stage.

Regarding your last points, we have reviewed the manuscript and introduced several modifications to soften our language regarding the generated cells and their clinical relevance:

Title: “full” was removed

Abstract, line 31: This sentence has been reworked and now points to new research perspectives towards therapeutic applications, supporting the idea that more research and optimization are needed before clinical applications can be considered.

New sentence: Thus, the presented thymic organoid model provides a practical platform for studying thymic cellular interactions and thymopoiesis in vitro, and opens new research perspectives towards cell-based therapies.

Line 517, “Thymopoiesis” was replaced with “T cell differentiation”: Importantly, we observed a CD3⁺ population arising at D17 and rapidly expanding at D21, reaching up to 60% of the hematopoietic population at D35, confirming active T cell differentiation in our system (Fig. 7a-b).

Line 654, “full” was removed from the sentence

Line 731, “confirming the ability of hTOs to support full thymopoiesis” was removed: Interestingly, flow cytometry confirmed the ability of our system to generate both $\alpha\beta$ and $\gamma\delta$ T lineages, in contrast to studies showing an abnormal restriction to $\gamma\delta$ T generation.

Line 740, “Thymopoiesis” was replaced with “T cell differentiation”: Given recent reports highlighting the critical role of intrathymic DCs in complementary thymocyte selection, this population may contribute to T cell differentiation in hTOs, potentially promoting commitment to the SP CD4⁺ fate.

Line 766: The passage “With the aforementioned necessary improvements and additional studies” was added at the beginning of the sentence: With the aforementioned necessary improvements and additional studies, we foresee two main clinical applications for this system.

Line 780: This sentence was reworked in the same manner as the abstract, shifting the focus towards new research perspectives rather than cell therapy applications.

Former version: The use of iPSc presents promising long-term perspectives for regenerative medicine and cellular therapies, particularly for the in vitro generation of engineered T lymphocytes.

New version: The use of iPSc opens promising research perspectives for long-term applications in regenerative medicine and cellular therapies, particularly via the in vitro generation of engineered T lymphocytes.